# MULTI-STEP GREEDY POLICIES IN MODEL-FREE DEEP REINFORCEMENT LEARNING

## ABSTRACT

Multi-step greedy policies have been extensively used in model-based Reinforcement Learning (RL) and in the case when a model of the environment is available (e.g., in the game of Go). In this work, we explore the benefits of multi-step greedy policies in model-free RL when employed in the framework of multi-step Dynamic Programming (DP): multi-step Policy and Value Iteration. These algorithms iteratively solve short-horizon decision problems and converge to the optimal solution of the original one. By using model-free algorithms as solvers of the short-horizon problems we derive fully model-free algorithms which are instances of the multi-step DP framework. As model-free algorithms are prone to instabilities w.r.t. the decision problem horizon, this simple approach can help in mitigating these instabilities and results in an improved model-free algorithms. We test this approach and show results on both discrete and continuous control problems.

## 1 INTRODUCTION

The field of Reinforcement learning (RL) span a wide variety of algorithms for solving decision-making problems through repeated interaction with the environment. By incorporating deep neural networks into RL algorithms, the field of RL has recently witnessed remarkable empirical success (e.g., Mnih et al. 2015; Lillicrap et al. 2015; Levine et al. 2016; Silver et al. 2017). Much of this success had been achieved by model-free RL algorithms, such as Q-learning and policy gradient. These algorithms are known to suffer from high variance in their estimations (Greensmith et al., 2004) and to have difficulties handling function approximation (e.g., Thrun & Schwartz 1993; Baird 1995; Van Hasselt et al. 2016; Lu et al. 2018). These problems are intensified in decision problems with long horizon, i.e., when the discount factor, $\gamma$, is large. Although using smaller values of $\gamma$ addresses the $\gamma$-dependent issues and leads to more stable algorithms (Petrik & Scherrer, 2009; Jiang et al., 2015), it comes with a cost, as the algorithm may return a *biased* solution, i.e., it may not converge to an optimal solution of the original decision problem (the one with large value of $\gamma$).

Efroni et al. (2018a) recently proposed another approach to mitigate the $\gamma$-dependant instabilities in RL in which they study a multi-step greedy versions of the well-known dynamic programming (DP) algorithms policy iteration (PI) and value iteration (VI) (Bertsekas & Tsitsiklis, 1996). Efroni et al. (2018a) also proposed an alternative formulation of the multi-step greedy policy, called $\kappa$-greedy policy, and studied the convergence of the resulted PI and VI algorithms: $\kappa$-PI and $\kappa$-VI. These two algorithms iteratively solve $\gamma\kappa$-discounted decision problems, whose reward has been shaped by the solution of the decision problem at the previous iteration. Unlike the *biased* solution obtained by solving the decision problem with a smaller value of $\gamma$, by iteratively solving decision problems with a smaller $\gamma\kappa$ horizon, the $\kappa$-PI and $\kappa$-VI algorithms could converge to an optimal policy of the original decision problem.

In this work, we derive and empirically validate model-free deep RL (DRL) implementations of $\kappa$-PI and $\kappa$-VI. In these implementations, we use DQN (Mnih et al., 2015) and TRPO (Schulman et al., 2015) for (approximately) solving $\gamma\kappa$-discounted decision problems (with shaped reward), which is the main component of the $\kappa$-PI and $\kappa$-VI algorithms. The experiments illustrate the performance of model-free algorithms can be improved by using them as solvers of multi-step greedy PI and VI schemes, as well as emphasize important implementation details while doing so.

## 2 PRELIMINARIES

In this paper, we assume that the agent's interaction with the environment is modeled as a discrete time $\gamma$-discounted Markov Decision Process (MDP), defined by $\mathcal{M}_\gamma = (\mathcal{S}, \mathcal{A}, P, R, \gamma, \mu)$, where $\mathcal{S}$ and $\mathcal{A}$ are the state and action spaces; $P \equiv P(s'|s,a)$ is the transition kernel; $R \equiv r(s,a)$

is the reward function with the maximum value of $R_{\max}$; $\gamma \in (0,1)$ is the discount factor; and $\mu$ is the initial state distribution. Let $\pi : \mathcal{S} \to \mathcal{P}(\mathcal{A})$ be a stationary Markovian policy, where $\mathcal{P}(\mathcal{A})$ is a probability distribution on the set $\mathcal{A}$. The value of $\pi$ in any state $s \in \mathcal{S}$ is defined as $V^\pi(s) \equiv \mathbb{E}[\sum_{t\geq 0} \gamma^t r(s_t, \pi(s_t))|s_0 = s, \pi]$, where the expectation is over all the randomness in policy, dynamics, and rewards. Similarly, the action-value function of $\pi$ is defined as $Q^\pi(s,a) = \mathbb{E}[\sum_{t\geq 0} \gamma^t r(s_t, \pi(s_t))|s_0 = s, a_0 = a, \pi]$. Since the rewards have the maximum value of $R_{\max}$, both $V$ and $Q$ functions have the maximum value of $V_{\max} = R_{\max}/(1-\gamma)$. An optimal policy $\pi^*$ is the policy with maximum value at every state. We call the value of $\pi^*$ the optimal value, and define it as $V^*(s) = \max_\pi \mathbb{E}[\sum_{t\geq 0} \gamma^t r(s_t, \pi(s_t))|s_0 = s, \pi]$, $\forall s \in \mathcal{S}$. Furthermore, we denote the state-action value of $\pi^*$ as $Q^*(s,a)$ and remind the following relation holds $V^*(s) = \max_a Q^*(s,a)$ for all $s$. The algorithms by which an is be solved (obtain an optimal policy) are mainly based on two popular DP algorithms: Policy Iteration (PI) and Value Iteration (VI). While VI relies on iteratively computing the optimal Bellman operator $T$ applied to the current value function $V$ (Eq. 1), PI relies on (iteratively) calculating a 1-step greedy policy $\pi_{\text{1-step}}$ w.r.t. to the value function of the current policy $V$ (Eq. 2):

$$(TV)(s) = \max_{a\in\mathcal{A}} \mathbb{E}[r(s_0, a) + \gamma V(s_1) \mid s_0 = s], \qquad \forall s \in \mathcal{S}, \tag{1}$$

$$\pi_{\text{1-step}}(s) \in \arg\max_{a\in\mathcal{A}} \mathbb{E}[r(s_0, a) + \gamma V(s_1) \mid s_0 = s], \quad \forall s \in \mathcal{S}. \tag{2}$$

It is known that $T$ is a $\gamma$-contraction w.r.t. the max norm and its unique fixed point is $V^*$, and the 1-step greedy policy w.r.t. $V^*$ is an optimal policy $\pi^*$. In practice, the state space is often large, and thus, we can only approximately compute Eqs. 1 and 2, which results in approximate PI (API) and VI (AVI) algorithms. These approximation errors then propagate through the iterations of the API and AVI algorithms. However, it has been shown that this (propagated) error can be controlled (Munos, 2003; 2005; Farahmand et al., 2010) and after $N$ steps, the algorithms approximately converge to a solution $\pi_N$ whose difference with the optimal value is bounded (see e.g., Scherrer 2014 for API):

$$\eta(\pi^*) - \eta(\pi_N) \leq C\delta/(1-\gamma)^2 + \gamma^N V_{\max}. \tag{3}$$

In Eq. 3, the scalar $\eta(\pi) = \mathbb{E}_{s\sim\mu}[V^\pi(s)]$ is the expected value function at the initial state,[1] $\delta$ represents the per-iteration error, and $C$ upper-bounds the mismatch between the sampling distribution and the distribution according to which the final value function is evaluated ($\mu$ in Eq. 3), and depends heavily on the dynamics. Finally, the second term on the RHS of Eq. 3 is the error due to initial values of policy/value, and decays with the number of iterations $N$.

## 3   $\kappa$-GREEDY POLICY & $\kappa$-PI AND $\kappa$-VI ALGORITHMS

---

**Algorithm 1** $\kappa$-Policy Iteration

1: **Initialize:** $\kappa \in [0,1]$, $\pi_0$, $N(\kappa)$
2: **for** $i = 0, 1, \ldots, N(\kappa) - 1$ **do**
3:   $V^{\pi_i} = \mathbb{E}[\sum_{t\geq 0} \gamma^t r_t \mid \pi_i]$
4:   $\pi_{i+1} \leftarrow \arg\max_\pi \mathbb{E}[\sum_{t=0}^{\infty} (\kappa\gamma)^t r_t(\kappa, V^{\pi_i})|\pi]$
5: **end for**
6: **Return** $\pi_{N(\kappa)}$

---

**Algorithm 2** $\kappa$-Value Iteration

1: **Initialize:** $\kappa \in [0,1]$, $V_0$, $N(\kappa)$
2: **for** $i = 0, 1, \ldots, N(\kappa) - 1$ **do**
3:   $V_{i+1} = \max_\pi \mathbb{E}[\sum_{t\geq 0} (\gamma\kappa)^t r_t(\kappa, V_i)|\pi]$
4: **end for**
5: $\pi_{N(\kappa)} \leftarrow \arg\max_\pi \mathbb{E}[\sum_{t\geq 0} (\kappa\gamma)^t r_t(\kappa, V_{N(\kappa)})|\pi]$
6: **Return** $\pi_{N(\kappa)}$

---

The optimal Bellman operator $T$ (Eq. 1) and 1-step greedy policy $\pi_{\text{1-step}}$ (Eq. 2) can be generalized to multi-step. The most straightforward form of this generalization is by replacing $T$ and $\pi_{\text{1-step}}$ with $h$-optimal Bellman operator and $h$-step greedy policy (i.e., a lookahead of horizon $h$) that are defined by substituting the 1-step return in Eqs. 1 and 2, $r(s_0, a) + \gamma V(s_1)$, with $h$-step return, $\sum_{t=0}^{h-1} r(s_t, a_t) + \gamma^h V(s_h)$, and computing the maximum over actions $a_0, \ldots, a_{h-1}$, instead of just $a_0$ (Bertsekas & Tsitsiklis, 1996). Efroni et al. (2018a) proposed an alternative form of multi-step optimal Bellman operator and multi-step greedy policy, called $\kappa$-optimal Bellman operator, $T_\kappa$, and $\kappa$-greedy policy, $\pi_\kappa$, for $\kappa \in [0,1]$, i.e.,

$$(T_\kappa V)(s) = \max_\pi \mathbb{E}[\sum_{t\geq 0} (\gamma\kappa)^t r_t(\kappa, V) \mid s_0 = s, \pi], \qquad \forall s \in \mathcal{S}, \tag{4}$$

$$\pi_\kappa(s) \in \arg\max_\pi \mathbb{E}[\sum_{t\geq 0} (\gamma\kappa)^t r_t(\kappa, V) \mid s_0 = s, \pi], \quad \forall s \in \mathcal{S}, \tag{5}$$

---

[1]Note that the LHS of Eq. 3 is the $\ell_1$-norm of $(V^{\pi^*} - V^{\pi_N})$ w.r.t. the initial state distribution $\mu$.

where the *shaped reward* $r_t(\kappa, V)$ w.r.t. the value function $V$ is defined as

$$r_t(\kappa, V) \equiv r(s_t, a_t) + (1 - \kappa)\gamma V(s_{t+1}). \tag{6}$$

It can be shown that the $\kappa$-greedy policy w.r.t. the value function $V$ is the optimal policy w.r.t. a $\kappa$-weighted geometric average of all future $h$-step returns (from $h = 0$ to $\infty$). This can be interpreted as TD($\lambda$) (Sutton & Barto, 2018) for policy improvement (see Efroni et al., 2018a, Sec. 6). The important difference is that TD($\lambda$) is used for policy evaluation and not for policy improvement.

From Eqs. 4 and 5, it is easy to see that solving these equations is equivalent to solving a surrogate $\gamma\kappa$-discounted MDP with the shaped reward $r_t(\kappa, V)$, which we denote by $\mathcal{M}_{\gamma\kappa}(V)$ throughout the paper. The optimal value of $\mathcal{M}_{\gamma\kappa}(V)$ (the surrogate MDP) is $T_\kappa V$ and its optimal policy is the $\kappa$-greedy policy, $\pi_\kappa$. Using the notions of $\kappa$-optimal Bellman operator, $T_\kappa$, and $\kappa$-greedy policy, $\pi_\kappa$, Efroni et al. (2018a) derived $\kappa$-PI and $\kappa$-VI algorithms, whose pseudocode is shown in Algorithms 1 and 2. $\kappa$-PI iteratively *(i)* evaluates the value of the current policy $\pi_i$, and *(ii)* set the new policy, $\pi_{i+1}$, to the $\kappa$-greedy policy w.r.t. the value of the current policy $V^{\pi_i}$, by solving Eq. 5. On the other hand, $\kappa$-VI repeatedly applies the $T_\kappa$ operator to the current value function $V_i$ (solves Eq. 4) to obtain the next value function, $V_{i+1}$, and returns the $\kappa$-greedy policy w.r.t. the final value $V_{N(\kappa)}$. Note that for $\kappa = 0$, the $\kappa$-greedy policy and $\kappa$-optimal Bellman operator are equivalent to their 1-step counterparts, defined by Eqs. 1 and 2, which indicates that $\kappa$-PI and $\kappa$-VI are generalizations of the seminal PI and VI algorithms.

It has been shown that both PI and VI converge to the optimal value with an exponential rate that depends on the discount factor $\gamma$, i.e., $\|V^* - V^{\pi_N}\|_\infty \leq O(\gamma^N)$ (see e.g., Bertsekas & Tsitsiklis, 1996; Scherrer, 2013). Analogously, Efroni et al. (2018a) showed that $\kappa$-PI and $\kappa$-VI converge with faster exponential rate of $\xi(\kappa) = \frac{\gamma(1-\kappa)}{1-\gamma\kappa} \leq \gamma$, i.e., $\|V^* - V^{\pi_{N(\kappa)}}\|_\infty \leq O(\xi(\kappa)^{N(\kappa)})$, with the cost that each iteration of these algorithms is computationally more expensive than that of PI and VI. Finally, we state the following two properties of $\kappa$-PI and $\kappa$-greedy policies that we use in our RL implementations of $\kappa$-PI and $\kappa$-VI algorithms in Sections 4 and 5:

**1)** *Asymptotic performance depends on $\kappa$.* The following bound that is similar to the one reported in Eq. 3 was proved by Efroni et al. (2018b, Thm. 5) for the performance of $\kappa$-PI:

$$\eta(\pi^*) - \eta(\pi_{N(\kappa)}) \leq \underbrace{C(\kappa)\delta(\kappa)/(1 - \gamma)^2}_{\text{Asymptotic Term}} + \underbrace{\xi(\kappa)^{N(\kappa)}V_{\max}}_{\text{Decaying Term}}, \tag{7}$$

where $\delta(\kappa)$ and $C(\kappa)$ are quantities similar to $\delta$ and $C$ in Eq. 3. Note that the first term on the RHS of Eq. 7 is independent of $N(\kappa)$, while the second one decays with $N(\kappa)$.

**2)** *Soft updates w.r.t. a $\kappa$-greedy policy does not necessarily improve the performance.* Let $\pi_\kappa$ be the $\kappa$-greedy policy w.r.t. $V^\pi$. Then, unlike for 1-step greedy policies, the performance of $(1-\alpha)\pi+\alpha\pi_\kappa$ (soft update) is not necessarily better than that of $\pi$ (Efroni et al., 2018b, Thm. 1). This hints that it would be advantages to use $\kappa$-greedy policies with 'hard' updates (using $\pi_\kappa$ as the new policy).

## 4    RL IMPLEMENTATIONS OF $\kappa$-PI AND $\kappa$-VI

As described in Sec. 3, implementing $\kappa$-PI and $\kappa$-VI requires iteratively solving a $\gamma\kappa$-discounted surrogate MDP with a shaped reward. If a model of the environment is given, the surrogate MDP can be solved using a DP algorithm (see Efroni et al., 2018a, Sec. 7). When the model is not available, it can be approximately solved by any model-free RL algorithm. In this paper, we focus on the case that the model is not available and propose RL implementations of $\kappa$-PI and $\kappa$-VI. The main question we investigate in this work is *how model-free RL algorithms should be implemented to efficiently solve the surrogate MDP in $\kappa$-PI and $\kappa$-VI.*

In this paper, we use DQN (Mnih et al., 2015) and TRPO (Schulman et al., 2015) as subroutines for estimating a $\kappa$-greedy policy (Line 4 in $\kappa$-PI, Alg. 1 and Line 5 in $\kappa$-VI, Alg. 2) or for estimating an optimal value of the surrogate MDP (Line 3 in $\kappa$-VI, Alg. 2). For estimating the value of the current policy (Line 3, in $\kappa$-PI, Alg. 1), we use standard policy evaluation deep RL (DRL) algorithms.

To implement $\kappa$-PI and $\kappa$-VI, we shall set the value of $N(\kappa) \in \mathbb{N}$, i.e., the total number of iterations of these algorithms, and determine the number of samples for each iteration. Since $N(\kappa)$ only appears in the second term of Eq. 7, an appropriate choice of $N(\kappa)$ is such that $C(\kappa)\delta(\kappa)/(1 - \gamma)^2 \simeq \xi(\kappa)^{N(\kappa)}V_{\max}$. Note that setting $N(\kappa)$ to a higher value would not dramatically improve the

---

**Algorithm 3** $\kappa$-PI-DQN

---

1: **Initialize** replay buffer $\mathcal{D}$, $Q$-networks $Q_\theta$, $Q_\phi$, and target networks $Q'_\theta$, $Q'_\phi$;
2: **for** $i = 0, \ldots, N(\kappa) - 1$ **do**
3:     # Policy Improvement
4:     **for** $t = 1, \ldots, T(\kappa)$ **do**
5:         Act by an $\epsilon$-greedy policy w.r.t. $Q_\theta(s_t, a)$, observe $r_t, s_{t+1}$, and store $(s_t, a_t, r_t, s_{t+1})$ in $\mathcal{D}$;
6:         Sample a batch $\{(s_j, a_j, r_j, s_{j+1})\}_{j=1}^N$ from $\mathcal{D}$;
7:         Update $\theta$ by DQN rule with $\{(s_j, a_j, r_j(\kappa, V_\phi), s_{j+1})\}_{j=1}^N$, where
8:             $V_\phi(s_{j+1}) = Q_\phi(s_{j+1}, \pi_{i-1}(s_{j+1}))$    and    $\pi_{i-1}(s) \in \arg\max_a Q'_\theta(s, a)$;
9:         Copy $\theta$ to $\theta'$ occasionally    $(\theta' \leftarrow \theta)$;
10:     **end for**
11:     # Policy Evaluation of $\pi_i(s) \in \arg\max_a Q'_\theta(s, a)$
12:     **for** $t = 1, \ldots, T(\kappa)$ **do**
13:         Sample a batch $\{(s_j, a_j, r_j, s_{j+1})\}_{j=1}^N$ from $\mathcal{D}$;
14:         Update $\phi$ by TD(0) off-policy rule with $\{(s_j, a_j, r_j, s_{j+1})\}_{j=1}^N$, and $\pi_i(s) \in \arg\max_a Q'_\theta(s, a)$;
15:         Copy $\phi$ to $\phi'$ occasionally    $(\phi' \leftarrow \phi)$;
16:     **end for**
17: **end for**

---

performance, because the asymptotic term in Eq. 7 is independent of $N(\kappa)$. In practice, since $\delta(\kappa)$ and $C(\kappa)$ are unknown, we set $N(\kappa)$ to satisfy the following equality:

$$\xi(\kappa)^{N(\kappa)} = C_{FA}, \tag{8}$$

where $C_{FA}$ is a hyper-parameter that depends on the *final-accuracy* we are aiming for. For example, if we expect the final accuracy being 90%, we would set $C_{FA} = 0.1$. Our results suggest that this approach leads to a reasonable choice for $N(\kappa)$, e.g., $N(\kappa = 0.99) \simeq 4$ and $N(\kappa = 0.5) \simeq 115$, for $C_{FA} = 0.1$ and $\gamma = 0.99$. As we increase $\kappa$, we expect less iterations are needed for $\kappa$-PI and $\kappa$-VI to converge to a good policy. Another important observation is that since the discount factor of the surrogate MDP that $\kappa$-PI and $\kappa$-VI solve at each iteration is $\gamma\kappa$, the *effective horizon* (the effective horizon of a $\gamma\kappa$-discounted MDP is $1/(1 - \gamma\kappa)$) of the surrogate MDP increases with $\kappa$.

Lastly, we need to determine the number of samples for each iteration of $\kappa$-PI and $\kappa$-VI. We allocate equal number of samples per iteration, denoted by $T(\kappa)$. Since the total number of samples, $T$, is known beforehand, we set the number of samples per iteration to

$$T(\kappa) = T/N(\kappa). \tag{9}$$

## 5   DQN AND TRPO IMPLEMENTATIONS OF $\kappa$-PI AND $\kappa$-VI

In this section, we study the use of DQN (Mnih et al., 2015) and TRPO (Schulman et al., 2015) in $\kappa$-PI and $\kappa$-VI algorithms. We first derive our DQN and TRPO implementations of $\kappa$-PI and $\kappa$-VI in Sections 5.1 and 5.2. We refer to the resulting algorithms as $\kappa$-PI-DQN, $\kappa$-VI-DQN, $\kappa$-PI-TRPO, and $\kappa$-VI-TRPO. It is important to note that for $\kappa = 1$, $\kappa$-PI-DQN and $\kappa$-VI-DQN are reduced to DQN, and $\kappa$-PI-TRPO and $\kappa$-VI-TRPO are reduced to TRPO. We then conduct a set of experiments with these algorithms, in Sections 5.1.1 and 5.2.1, in which we carefully study the effect of $\kappa$ and $N(\kappa)$ (or equivalently the hyper-parameter $C_{FA}$, defined by Eq. 8) on their performance. In these experiments, we specifically focus on answering the following questions:

1. Is the performance of DQN and TRPO improve when using them as $\kappa$-greedy solvers in $\kappa$-PI and $\kappa$-VI? Is there a performance tradeoff w.r.t. to $\kappa$?

2. Following $\kappa$-PI and $\kappa$-VI, our DQN and TRPO implementations of these algorithms devote a significant number of sample $T(\kappa)$ to each iteration. Is this needed or a 'naive' choice of $T(\kappa) = 1$, or equivalently $N(\kappa) = T$, works just well, for all values of $\kappa$?

### 5.1   DQN IMPLEMENTATION OF $\kappa$-PI AND $\kappa$-VI

Algorithm 3 contains the pseudo-code of $\kappa$-PI-DQN. Due to space constraints, we report its detailed pseudo-code in Appendix A.1 (Alg. 5). In the *policy improvement stage* of $\kappa$-PI-DQN, we use DQN to solve the $\gamma\kappa$-discounted surrogate MDP with the shaped reward $r_t(\kappa, V_\phi \simeq V^{\pi_{i-1}})$,

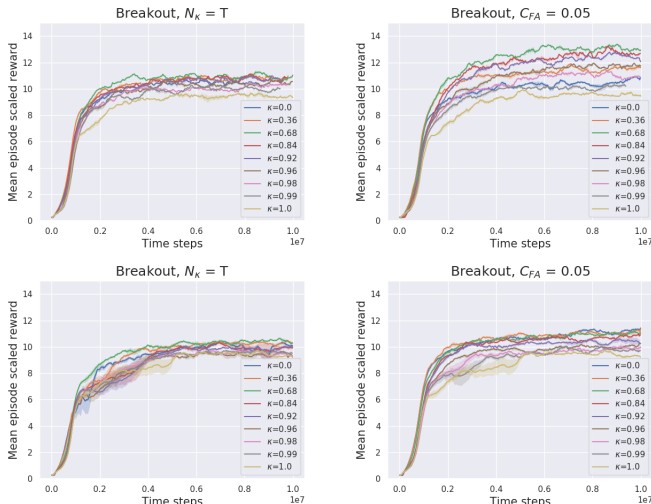

Figure 1: Training performance of $\kappa$-PI-DQN (Top) and $\kappa$-VI-DQN (Bottom) on Breakout, for the hyper-parameter $C_{FA} = 0.05$ (right) and for the 'naive' baseline $N(\kappa) = T$ (left).

i.e., at the end of this stage $\mathcal{M}_{\gamma\kappa}(V_\phi)$. The output of the DQN is approximately the optimal $Q$-function of $\mathcal{M}_{\gamma\kappa}(V_\phi)$, and thus, the $\kappa$-greedy policy w.r.t. $V_\phi$ is equal to $\arg\max_a Q_\theta(\cdot, a)$. At the *policy evaluation stage*, we use off-policy TD(0) to evaluate the $Q$-function of the current policy $\pi_i$, i.e., $Q_\phi \simeq Q^{\pi_i}$. Although what is needed on Line 8 is an estimate of the value function of the current policy, $V_\phi \simeq V^{\pi_{i-1}}$, we chose to evaluate the $Q$-function of $\pi_i$: the data in our disposal (the transitions stored in the replay buffer) is an off-policy data and the $Q$-function of a fixed policy can be easily evaluated with this type of a data using off-policy TD(0), unlike the value function.

**Remark 1** *In order for $V_\phi$ to be an accurate estimate of the value function of $\pi_{i-1}$ on Line 8, we should use an additional target network, $\widetilde{Q}'_\theta$, that remains unchanged during the policy improvement stage. This network should be used in $\pi_{i-1}(\cdot) = \arg\max_a \widetilde{Q}'_\theta(\cdot, a)$ on Line 8, and be only updated right after the improvement stage on Line 11. However, to reduce the space complexity of the algorithm, we do not use this additional target network and compute $\pi_{i-1}$ on Line 8 as $\arg\max Q'_\theta$, despite the fact that $Q'_\theta$ changes during the improvement stage.*

We report the pseudo-code of $\kappa$-VI-DQN in Appendix A.1 (Alg. 6). Note that $\kappa$-VI simply repeats $V \leftarrow T_\kappa V$ and computes $T_\kappa V$, which is the optimal value of the surrogate MDP $\mathcal{M}_{\gamma\kappa}(V)$. In $\kappa$-VI-DQN, we repeatedly solve $\mathcal{M}_{\gamma\kappa}(V)$ by DQN, and use its optimal $Q$-function to shape the reward of the next iteration. Let $Q^*_{\gamma\kappa,V}$ and $V^*_{\gamma\kappa,V}$ be the optimal $Q$ and $V$ functions of $\mathcal{M}_{\gamma\kappa}(V)$. Then, we have $\max_a Q^*_{\gamma\kappa,V}(s,a) = V^*_{\gamma\kappa,V}(s) = (T_\kappa V)(s)$, where the first equality is by definition (Sec. 2) and the second one holds since $T_\kappa V$ is the optimal value of $\mathcal{M}_{\gamma\kappa}(V)$ (Sec. 3). Therefore, in $\kappa$-VI-DQN, we shape the reward of each iteration by $\max_a Q_\phi(s,a)$, where $Q_\phi$ is the output of the DQN from the previous iteration, i.e., $\max_a Q_\phi(s,a) \simeq T_\kappa V_{i-1}$.

### 5.1.1 $\kappa$-PI-DQN AND $\kappa$-VI-DQN EXPERIMENTS

In this section, we empirically analyze the performance of the $\kappa$-PI-DQN and $\kappa$-VI-DQN algorithms on the Atari domains: Breakout, Seaquest, SpaceInvaders, and Enduro (Bellemare et al., 2013). We start by performing an ablation test on three values of parameter $C_{FA} = \{0.001, 0.05, 0.2\}$ on the Breakout domain. The value of $C_{FA}$ sets the number of samples per iteration $T(\kappa)$ (Eq. 8) and the total number of iterations $N(\kappa)$ (Eq. 9). Aside from $C_{FA}$, we set the total number of samples to $T \simeq 10^6$. This value represents the number of samples after which our DQN-based algorithms approximately converge. For each value of $C_{FA}$, we test $\kappa$-PI-DQN and $\kappa$-VI-DQN for several $\kappa$ values. In both algorithms, the best performance was obtained with $C_{FA} = 0.05$, thus, we set $C_{FA} = 0.05$ in our experiments with other Atari domains.

| Domain | Alg. | $\kappa_{\text{best}}$ | $\kappa = 0$ | DQN, $\kappa = 1$ | $N(\kappa) = T, \kappa_{\text{best}}$ |
|---|---|---|---|---|---|
| Breakout | $\kappa$-PI | $224_{(\pm 5)}$, $\kappa$=0.68 | $160_{(\pm 3)}$ | $131_{(\pm 3)}$ | $171_{(\pm 1)}$, $\kappa$=0.68 |
| | $\kappa$-VI | $180_{(\pm 5)}$, $\kappa$=0.68 | $179_{(\pm 6)}$ | | |
| SpaceInv. | $\kappa$-PI | $747_{(\pm 23)}$, $\kappa$=0.84 | $611_{(\pm 15)}$ | $685_{(\pm 24)}$ | $695_{(\pm 16)}$, $\kappa$=0.92 |
| | $\kappa$-VI | $707_{(\pm 32)}$, $\kappa$=0.36 | $669_{(\pm 16)}$ | | |
| Seaquest | $\kappa$-PI | $5159_{(\pm 509)}$, $\kappa$=0.84 | $2732_{(\pm 281)}$ | $3207_{(\pm 248)}$ | $4371_{(\pm 466)}$, $\kappa$=0.84 |
| | $\kappa$-VI | $3394_{(\pm 86)}$, $\kappa$=0.36 | $2631_{(\pm 496)}$ | | |
| Enduro | $\kappa$-PI | $544_{(\pm 29)}$, $\kappa$=0.84 | $371_{(\pm 215)}$ | $355_{(\pm 52)}$ | $547_{(\pm 17)}$, $\kappa$=0.68 |
| | $\kappa$-VI | $499_{(\pm 18)}$, $\kappa$=0.84 | $492_{(\pm 28)}$ | | |
| BeamRider | $\kappa$-PI | $3968_{(\pm 78)}$, $\kappa$=1.0 | $3654_{(\pm 778)}$ | $3968_{(\pm 78)}$ | $3968_{(\pm 78)}$, $\kappa$=1.0 |
| | $\kappa$-VI | $4077_{(\pm 303)}$, $\kappa$=0.68 | $4052_{(\pm 462)}$ | | |
| Qbert | $\kappa$-PI | $8276_{(\pm 202)}$, $\kappa$=0.36 | $6900_{(\pm 149)}$ | $7322_{(\pm 280)}$ | $8042_{(\pm 442)}$, $\kappa$=0.92 |
| | $\kappa$-VI | $7924_{(\pm 267)}$, $\kappa$=0.68 | $7585_{(\pm 587)}$ | | |

Table 1: The final training performance of $\kappa$-PI-DQN and $\kappa$-VI-DQN on the Atari domains, for the hyper-parameter $C_{FA} = 0.05$. The values represent the empirical mean $\pm$ empirical standard deviation.

Figure 1 shows the training performance of $\kappa$-PI-DQN (Top) and $\kappa$-VI-DQN (Bottom) for the best value of $C_{FA} = 0.05$, as well as for the 'naive' baseline $T(\kappa) = 1$, or equivalently $N(\kappa) = T$, on Breakout. The results on Breakout for the other values of $C_{FA}$ and the results on the other Atari domains for $C_{FA} = 0.05$ have been reported in Appendix A.2. Table 1 shows the final training performance of $\kappa$-PI-DQN and $\kappa$-VI-DQN on the Atari domains with $C_{FA} = 0.05$. Note that the scores reported in Table 1 are the actual returns of the Atari domains, while the vertical axis in the plots of Figure 1 corresponds to a scaled return. We plot the scaled return, since this way it would be easier to reproduce our results using the OpenAI Baselines codebase (Hill et al., 2018).

The results of Fig. 1 and Table 1, as well as those in Appendix A.2, exhibit that both $\kappa$-PI-DQN and $\kappa$-VI-DQN improve the performance of DQN ($\kappa = 1$). Moreover, they show that setting $N(\kappa) = T$ leads to a clear degradation of the final training performance on all of the domains expect Enduro, which attains better performance for $N(\kappa) = T$. Although the performance degrades, the results for $N(\kappa) = T$ are still better than for DQN.

## 5.2 TRPO IMPLEMENTATION OF $\kappa$-PI AND $\kappa$-VI

Algorithm 4 contains the pseudo-code of $\kappa$-PI-TRPO (detailed pseudo-code in Appendix A.1). TRPO iteratively updates the current policy using its return and an estimate of its value function. In our $\kappa$-PI-TRPO, at each iteration $i$: **1)** we use the estimate of the current policy $V'_\phi \simeq V^{\pi_{i-1}}$ (computed in the previous iteration) to calculate the return $R(\kappa, V'_\phi)$ and an estimate of the value function $V_\theta$ of the surrogate MDP $\mathcal{M}_{\gamma\kappa}(V'_\phi)$, **2)** we use the return $R(\kappa, V'_\phi)$ and $V_\theta$ to compute the new policy $\pi_i$, and **3)** we estimate the value of the new policy $V_\phi \simeq V^{\pi_i}$ on the original, $\gamma$ discounted, MDP.

In Appendix B.1 we provide the pseudocode of $\kappa$-VI-TRPO derived by the $\kappa$-VI meta algorithm. As previously noted, $\kappa$-VI iteratively solves the $\gamma\kappa$ discounted surrogate MDP and uses its optimal value $T_\kappa V_{i-1}$ to shape the reward of the surrogated MDP in the i'th iteration. With that in mind, consider $\kappa$-PI-TRPO. Notice that as $\pi_\theta$ converges to the optimal policy of the surrogate $\gamma\kappa$ discounted MDP, $V_{\bar{\theta}}$ converges to the optimal value of the surrogate MDP, i.e., it converges to $T_\kappa V_{i-1} = T_\kappa V_{i-1}$. Thus, $\kappa$-PI-TRPO can be turn to $\kappa$-VI-TRPO by eliminating the policy evaluation stage, and simply copy $\phi \leftarrow \bar{\theta}$, meaning, $V_\phi \leftarrow V_{\bar{\theta}} = T_\kappa V_\phi$.

### 5.2.1 $\kappa$-PI-TRPO AND $\kappa$-VI-TRPO EXPERIMENTS

In this section, we empirically analyze the performance of the $\kappa$-PI-TRPO and $\kappa$-VI-TRPO algorithms on the MuJoCo domains: Walker2d-v2, Ant-v2, HalfCheetah-v2, HumanoidStandup-v2, and Swimmer-v2, (Todorov et al., 2012). As in Section 5.1.1, we start by performing an ablation test on the parameter $C_{FA} = \{0.001, 0.05, 0.2\}$ on the Walker domain. We set the total number of iterations to 2000, with each iteration consisting 1000 samples. Thus, the total number of samples is $T \simeq 2 \times 10^6$. This is the number of samples after which our TRPO-based algorithms approximately converge. For each value of $C_{FA}$, we test $\kappa$-PI-TRPO and $\kappa$-VI-TRPO for several $\kappa$ values. In both algorithms, the best performance was obtained with $C_{FA} = 0.2$, thus, we set $C_{FA} = 0.2$ in our experiments with other MuJoCo domains.

---

**Algorithm 4** $\kappa$-PI-TRPO

---

1: **Initialize** $V$-networks $V_\theta$ and $V_\phi$, policy network $\pi_\psi$, and target network $V'_\phi$;
2: **for** $i = 0, \ldots, N(\kappa) - 1$ **do**
3:    **for** $t = 1, \ldots, T(\kappa)$ **do**
4:       Simulate the current policy $\pi_\psi$ for $M$ steps and calculate the following two returns for all steps $j$:
5:          $R_j(\kappa, V'_\phi) = \sum_{t=j}^{M} (\gamma\kappa)^{t-j} r_t(\kappa, V'_\phi)$     and     $\rho_j = \sum_{t=j}^{M} \gamma^{t-j} r_t$;
6:       Update $\theta$ by minimizing the batch loss function:     $\mathcal{L}_{V_\theta} = \frac{1}{N} \sum_{j=1}^{N} (V_\theta(s_j) - R_j(\kappa, V'_\phi))^2$;
7:       # Policy Improvement
8:       Update $\psi$ using TRPO by the batch $\{(R_j(\kappa, V'_\phi), V_\theta(s_j))\}_{j=1}^{N}$;
9:       # Policy Evaluation
10:      Update $\phi$ by minimizing the batch loss function:     $\mathcal{L}_{V_\phi} = \frac{1}{N} \sum_{j=1}^{N} (V_\phi(s_j) - \rho_j)^2$;
11:    **end for**
12:    Copy $\phi$ to $\phi'$   $(\phi' \leftarrow \phi)$;
13: **end for**

---

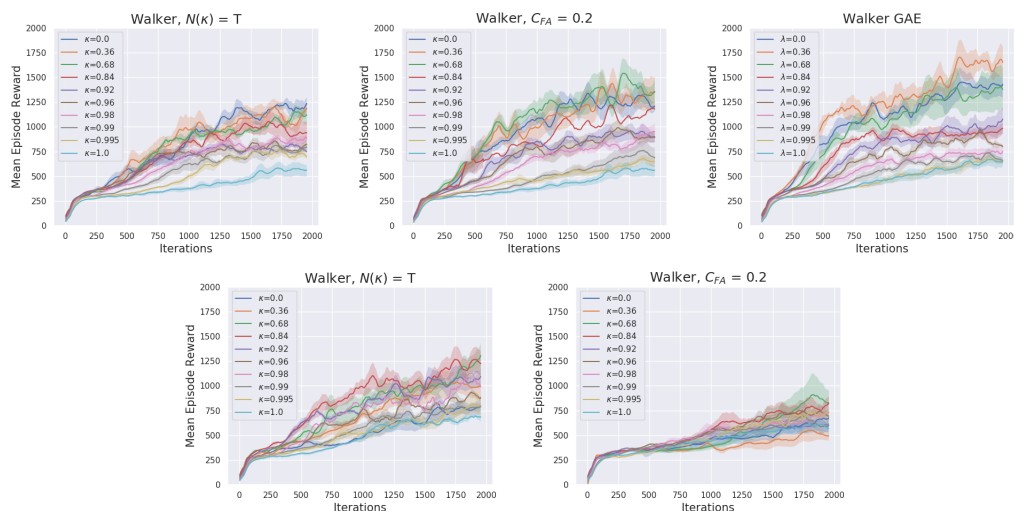

Figure 2: Training performance of $\kappa$-PI-TRPO (Top) and $\kappa$-VI-TRPO (Bottom) on Walker, for the hyper-parameter $C_{FA} = 0.2$ (right) and for the 'naive' baseline $N(\kappa) = T$ (left).

Figure 2 shows the training performance of $\kappa$-PI-TRPO (Top) and $\kappa$-VI-TRPO (Bottom) for the best value of $C_{FA} = 0.2$, as well as for the 'naive' baseline $T(\kappa) = 1$, or equivalently $N(\kappa) = T$, on Walker. The results on Walker for the other $C_{FA}$ values and the other MuJoCo domains for $C_{FA} = 0.2$ have been reported in Appendix B.3. Table 2 shows the final training performance of $\kappa$-PI-TRPO and $\kappa$-VI-TRPO on the MuJoCo domains with $C_{FA} = 0.2$.

The results of Figure 2 and Table 2, as well as those in Appendix B.3, exhibit that both $\kappa$-PI-TRPO and $\kappa$-VI-TRPO yield better performance than TRPO ($\kappa = 1$). Furthermore, they show that the algorithms with $C_{FA} = 0.2$ perform better than with $N(\kappa) = T$. However, the improvement is less significant relative to the DQN-based results in Section 5.1.1.

### 5.2.2 COMPARISON WITH THE GENERALIZED ADVANTAGE ESTIMATION ALGORITHM

There is an intimate relation between $\kappa$-PI and the GAE algorithm Schulman et al. (2016) which we elaborate on in this section. In GAE the policy is updated by the gradient:

$$\nabla_\theta \mathbb{E}_{s \sim \mu}[V^{\pi_\theta}(s)] = \mathbb{E}_{s_0 \sim d_{\mu,\pi}} \left[ \nabla_\theta \log \pi_\theta(s_0) \sum_t (\gamma\lambda)^t \delta(V) \right]; \; \delta(V) = r_t + \gamma V_{t+1} - V_t, \quad (10)$$

which can be interpreted as a gradient step in a $\gamma\lambda$ discounted MDP with rewards $\delta(V)$, which we refer here as $\mathcal{M}_{\gamma\lambda}^{\delta(V)}$. As noted in Efroni et al. (2018a), Section 6, the optimal policy of the MDP $\mathcal{M}_{\gamma\lambda}^{\delta(V)}$ is the optimal policy of $\mathcal{M}_{\gamma\kappa}(V)$ with $\kappa = \lambda$, i.e., the $\kappa$-greedy policy w.r.t. $V$: thus, the

| Domain | Alg. | $\kappa_{\text{best}}$ | $\kappa=0$ | TRPO, $\kappa=1$ | $N(\kappa)=T, \kappa_{\text{best}}$ | GAE |
|---|---|---|---|---|---|---|
| Walker | $\kappa$-PI | 1352(±233), $\kappa$=0.68 | 1205 (±99) | 560 (±117) | 1158(±75), $\kappa$=0.36 | 1664(±318), $\lambda$=0.36 |
| | $\kappa$-VI | 827(±269), $\kappa$=0.68 | 669(±125) | | | |
| Ant | $\kappa$-PI | 1359(±326), $\kappa$=0.68 | 1083(±205) | -18.47(±2) | 1225(±141), $\kappa$=0.0 | 1152(±255), $\lambda$=0.0 |
| | $\kappa$-VI | 2916(±455), $\kappa$=0.68 | 1809(±342) | | | |
| HalfCheetah | $\kappa$-PI | 1367(±406), $\kappa$=0.36 | 855(±160) | 74(±202) | 1450(±200), $\kappa$=0.36 | 1453(±203), $\lambda$=0.36 |
| | $\kappa$-VI | 1735(±800), $\kappa$=0.36 | 1078(±48) | | | |
| HumanoidStand | $\kappa$-PI | 73743(±1988), $\kappa$=0.99 | 73486(±1211) | 67545(±1545) | 72588(±1929), $\kappa$=0.98 | 71420(±1401), $\lambda$=0.98 |
| | $\kappa$-VI | 74063(±1779), $\kappa$=0.99 | 51323(±1805) | | | |
| Swimmer | $\kappa$-PI | 108(±17), $\kappa$=1.0 | 43(±3) | 108(±17) | 108(±17), $\kappa$=1.0 | 108(±14), $\lambda=1.0$ |
| | $\kappa$-VI | 108(±17), $\kappa$=1.0 | 46(±1) | | | |
| Hopper | $\kappa$-PI | 1872(±191), $\kappa$=0.68 | 805(±291) | 1193(±353) | 1491(±157), $\kappa$=0.96 | 1745(±300), $\lambda=0.68$ |
| | $\kappa$-VI | 1043(±95), $\kappa$=0.92 | 590(±246) | | | |

Table 2: The final training performance of $\kappa$-PI-TRPO and $\kappa$-VI-TRPO on the MuJoCo domains, for the hyper-parameter $C_{FA} = 0.2$. The values represent the empirical mean ± empirical standard deviation.

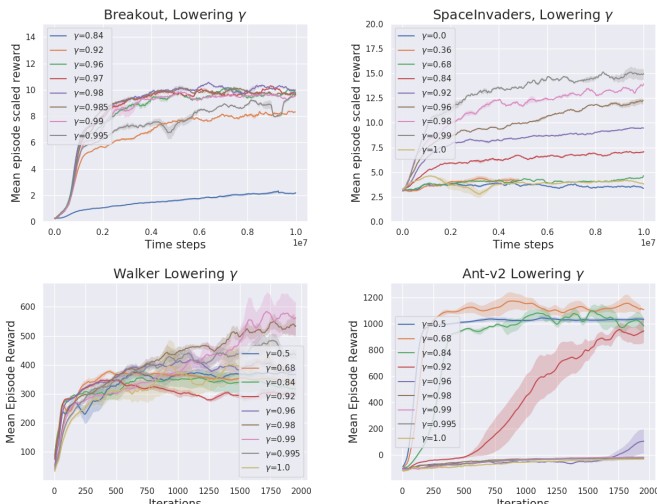

Figure 3: Lowering the discount factor

optimal policy of $\mathcal{M}_{\gamma\lambda}^{\delta(V)}$ is the $\kappa$-greedy policy w.r.t. $V$. GAE, instead of solving the $\kappa$-greedy policy while keeping $V$ fixed, changes the policy and updates $V$ by the return concurrently. Thus, this approach is conceptually similar to $\kappa$-PI-TRPO with $N(\kappa) = T$. There, the value and policy are concurrently updated as well, without clear separation between the update of the policy and the value.

In Figure 2 and Table 2 the performance of GAE is compared to the one of $\kappa$-PI-TRPO and $\kappa$-VI-TRPO. The performance of the latter two is slightly better than the one of GAE.

**Remark 2 (Implementation of GAE)** *We used the OpenAI baseline implementation of GAE with a small modification. In the baseline code, the value network is updated w.r.t. to the target $\sum_t (\gamma\lambda)^t r_t$, whereas in Schulman et al. (2016) the authors used the target $\sum_t \gamma^t r_t$ (see Schulman et al. (2016), Eq.28). We chose the latter form in our implementation to be in accord with Schulman et al. (2016).*

### 5.3 DQN AND TRPO PERFORMANCE VERSUS THE DISCOUNT FACTOR

To supply with a more complete view on our experiments, we tested the performance of the "vanilla" DQN and TRPO when trained with different $\gamma$ values than the previously used one ($\gamma = 0.99$). As evident in Figure 3, only for the Ant domain this approach resulted in improved performance when for TRPO trained with $\gamma = 0.68$. It is interesting to observe that for the Ant domain the performance of $\kappa$-PI-TRPO and especially of $\kappa$-VI-TRPO (Table 2) significantly surpassed the one of TRPO

trained with $\gamma = 0.68$. The performance of DQN and TRPO on the Breakout, SpaceInvaders and Walker domains decreased or remained unchanged in the tested $\gamma$ values. Thus, on these domains, changing the discount factor does not improve the DQN and TRPO algorithms, as using $\kappa$-PI or $\kappa$-VI with smaller $\kappa$ value do.

It is interesting to observe that the performance on the Mujoco domains for small $\gamma$, e.g., $\gamma = 0.68$, achieved good performance, whereas for the Atari domains the performance degraded with lowering $\gamma$. This fits the nature of these domains: in the Mujoco domains the decision problem inherently has much shorter horizon than in the Atari domains.

Furthermore, it is important to stress that $\gamma$ and $\kappa$ are two different parameters an algorithm designer may use. For example, one can perform a scan of $\gamma$ value, fix $\gamma$ to the one with optimal performance, and then test the performance of different $\kappa$ values.

## 6   Discussion and Future Work

In this work we formulated and empirically tested simple generalizations of DQN and TRPO derived by the theory of multi-step DP and, specifically, of $\kappa$-PI and $\kappa$-VI algorithms. The empirical investigation reveals several points worth emphasizing.

1. *$\kappa$-PI is better than $\kappa$-VI for the Atari domains.* In most of the experiments on the Atari domains $\kappa$-PI-DQN has better performance than $\kappa$-VI-DQN. This might be expected as the former uses extra information not used by the latter: $\kappa$-PI estimates the value of current policy whereas $\kappa$-VI ignores this information.

2. *For the Gym domains $\kappa$-VI performs slightly better than $\kappa$-PI.* For the Gym domains $\kappa$-VI-TRPO performs slightly better than $\kappa$-PI-TRPO. We conjecture that the reason for the discrepancy relatively to the Atari domains lies in the inherent structure of the tasks of the Gym domains: they are inherently short horizon decision problems. For this reason, the problems can be solved with smaller discount factor (as empirically demonstrated in Section 5.3) and information on the policy's value is not needed.

3. *Non trivial $\kappa$ value improves the performance.* In the vast majority of our experiments both $\kappa$-PI and $\kappa$-VI improves over the performance of their vanilla counterparts (i.e., $\kappa = 1$), except for the Swimmer and BeamRider domains from Mujoco and Atari suites. Importantly, the performance of the algorithms was shown to be 'smooth' in the parameter $\kappa$. This suggests careful hyperparameter tuning of $\kappa$ is not of great necessity.

4. *Using the 'naive' choice of $N(\kappa) = T$ deteriorates the performance.* Choosing the number of iteration by Eq. 8 improves the performance on the tested domains.

An interesting future work would be to test model-free algorithms which use other variants of greedy policies (Bertsekas & Tsitsiklis, 1996; Bertsekas, 2018; Efroni et al., 2018a; Sun et al., 2018; Shani et al., 2019). Furthermore, and although in this work we focused on model-free DRL, it is arguably more natural to use multi-step DP in model-based DRL (e.g.,Kumar et al., 2016; Talvitie, 2017; Luo et al., 2018; Janner et al., 2019). Taking this approach, the multi-step greedy policy would be solved with an approximate model. We conjecture that in this case one may set $\kappa$ – or more generally, the planning horizon – as a function of the approximate model's 'quality': as the approximate model gets closer to the real model larger $\kappa$ can be used. We leave investigating such relation in theory and practice to future work. Lastly, an important next step in continuation to our work is to study algorithms with an adaptive $\kappa$ parameter. This, we believe, would greatly improve the resulting methods, and possibly be done by studying the relation between the different approximation errors (i.e., errors in gradient and value estimation, Ilyas et al., 2018), the performance and the $\kappa$ value that should be used by the algorithm.

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

# A    DQN IMPLEMENTATION OF $\kappa$-PI AND $\kappa$-VI

## A.1    PSEUDO-CODES

In this section, we report the detailed pseudo-codes of the $\kappa$-PI-DQN and $\kappa$-VI-DQN algorithms, described in Section 5.1, side-by-side.

---

**Algorithm 5** $\kappa$-PI-DQN

---

1:  **Initialize** replay buffer $\mathcal{D}$, and $Q$-networks $Q_\theta$ and $Q_\phi$ with random weights $\theta$ and $\phi$;
2:  **Initialize** target networks $Q'_\theta$ and $Q'_\phi$ with weights $\theta' \leftarrow \theta$  and  $\phi' \leftarrow \phi$;
3:  **for**  $i = 0, \ldots, N(\kappa) - 1$ **do**
4:      # Policy Improvement
5:      **for** $t = 1, \ldots, T(\kappa)$ **do**
6:          Select $a_t$ as an $\epsilon$-greedy action w.r.t. $Q_\theta(s_t, a)$;
7:          Execute $a_t$, observe $r_t$ and $s_{t+1}$, and store the tuple $(s_t, a_t, r_t, s_{t+1})$ in $\mathcal{D}$;
8:          Sample a random mini-batch $\{(s_j, a_j, r_j, s_{j+1})\}_{j=1}^N$ from $\mathcal{D}$;
9:          Update $\theta$ by minimizing the following loss function:
10:             $\mathcal{L}_{\mathcal{Q}_\theta} = \frac{1}{N} \sum_{j=1}^N \left( Q_\theta(s_j, a_j) - (r_j(\kappa, V_\phi) + \gamma\kappa \max_a Q'_\theta(s_{j+1}, a)) \right)^2$,     where
11:             $V_\phi(s_{j+1}) = Q_\phi(s_{j+1}, \pi_{i-1}(s_{j+1}))$    and    $\pi_{i-1}(s_{j+1}) \in \arg\max_a Q'_\theta(s_{j+1}, a)$;
12:          Copy $\theta$ to $\theta'$ occasionally    $(\theta' \leftarrow \theta)$;
13:      **end for**
14:      # Policy Evaluation
15:      Set $\pi_i(s) \in \arg\max_a Q'_\theta(s, a)$;
16:      **for** $t' = 1, \ldots, T(\kappa)$ **do**
17:          Sample a random mini-batch $\{(s_j, a_j, r_j, s_{j+1})\}_{j=1}^N$ from $\mathcal{D}$;
18:          Update $\phi$ by minimizing the following loss function:
19:             $\mathcal{L}_{\mathcal{Q}_\phi} = \frac{1}{N} \sum_{j=1}^N \left( Q_\phi(s_j, a_j) - (r_j + \gamma Q'_\phi(s_{j+1}, \pi_i(s_{j+1}))) \right)^2$;
20:          Copy $\phi$ to $\phi'$ occasionally    $(\phi' \leftarrow \phi)$;
21:      **end for**
22:  **end for**

---

---

**Algorithm 6** $\kappa$-VI-DQN

---

1:  **Initialize** replay buffer $\mathcal{D}$, and $Q$-networks $Q_\theta$ and $Q_\phi$ with random weights $\theta$ and $\phi$;
2:  **Initialize** target network $Q'_\theta$ with weights  $\theta' \leftarrow \theta$;
3:  **for**  $i = 0, \ldots, N(\kappa) - 1$ **do**
4:      # Evaluate $T_\kappa V_\phi$ and the $\kappa$-greedy policy w.r.t. $V_\phi$
5:      **for** $t = 1, \ldots, T(\kappa)$ **do**
6:          Select $a_t$ as an $\epsilon$-greedy action w.r.t. $Q_\theta(s_t, a)$;
7:          Execute $a_t$, observe $r_t$ and $s_{t+1}$, and store the tuple $(s_t, a_t, r_t, s_{t+1})$ in $\mathcal{D}$;
8:          Sample a random mini-batch $\{(s_j, a_j, r_j, s_{j+1})\}_{j=1}^N$ from $\mathcal{D}$;
9:          Update $\theta$ by minimizing the following loss function:
10:             $\mathcal{L}_{\mathcal{Q}_\theta} = \frac{1}{N} \sum_{j=1}^N \left( Q_\theta(s_j, a_j) - (r_j(\kappa, V_\phi) + \kappa\gamma \max_a Q'_\theta(s_{j+1}, a)) \right)^2$,     where
11:             $V_\phi(s_{j+1}) = Q_\phi(s_{j+1}, \pi(s_{j+1}))$    and    $\pi(s_{j+1}) \in \arg\max_a Q'_\theta(s_{j+1}, a)$;
12:          Copy $\theta$ to $\theta'$ occasionally    $(\theta' \leftarrow \theta)$;
13:      **end for**
14:      Copy $\theta$ to $\phi$   $(\phi \leftarrow \theta)$
15:  **end for**

---

| Hyperparameter | Value |
|---|---|
| Horizon (T) | 1000 |
| Adam stepsize | $1 \times 10^{-4}$ |
| Target network update frequency | 1000 |
| Replay memory size | 100000 |
| Discount factor | 0.99 |
| Total training time steps | 10000000 |
| Minibatch size | 32 |
| Initial exploration | 1 |
| Final exploration | 0.1 |
| Final exploration frame | 1000000 |
| #Runs used for plot averages | 4 |
| Confidence interval for plot runs | $\sim 70\%$ |

Table 3: Hyperparameters for $\kappa$-PI-DQN and $\kappa$-VI-DQN.

## A.2 ABLATION TEST FOR $C_{FA}$

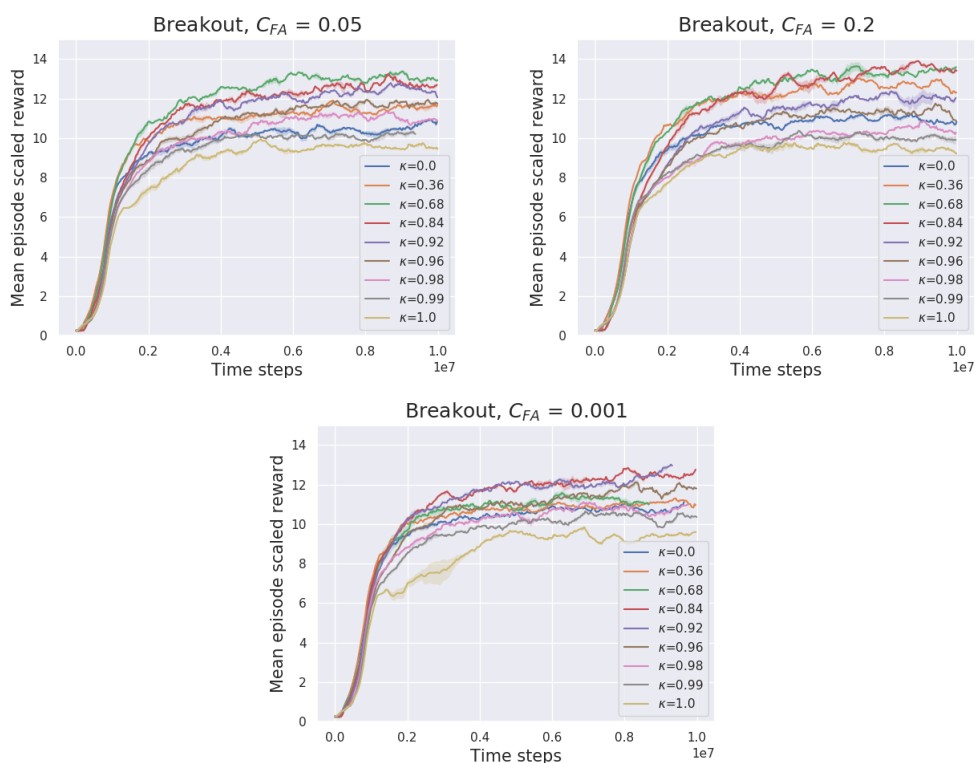

Figure 4: Performance of $\kappa$-PI-DQN and $\kappa$-VI-DQN on Breakout for different values of $C_{FA}$.

## A.3 $\kappa$-PI-DQN AND $\kappa$-VI-DQN PLOTS

In this section, we report additional results of the application of $\kappa$-PI-DQN and $\kappa$-VI-DQN on the Atari domains. A summary of these results has been reported in Table 1 in the main paper.

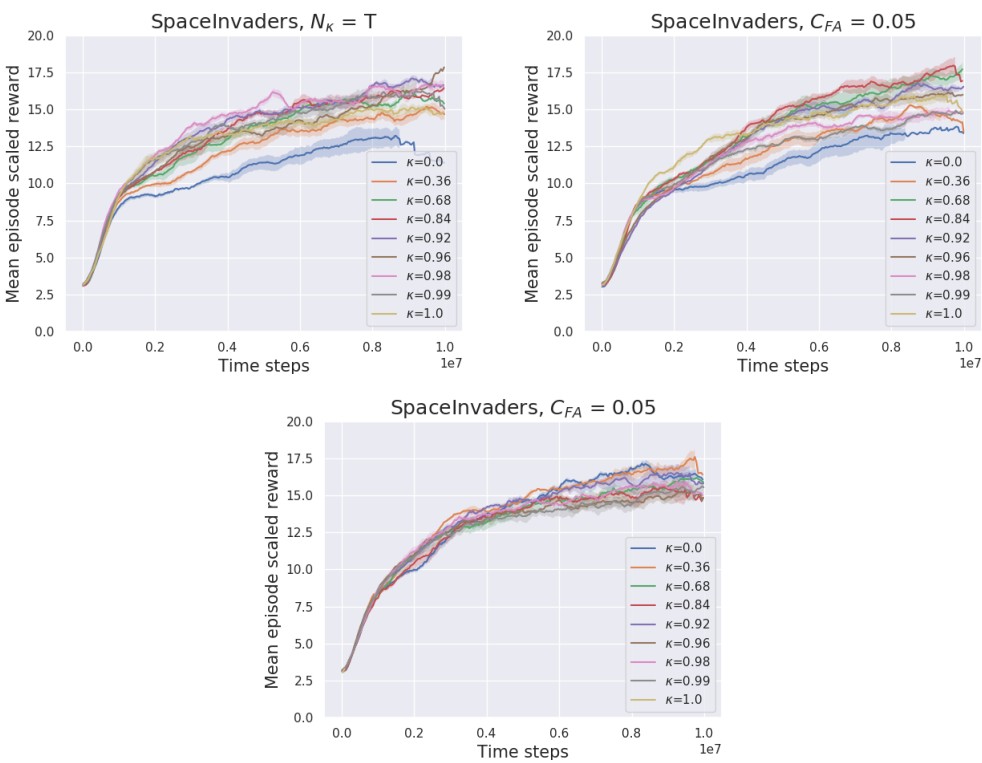

Figure 5: Training performance of $\kappa$-PI-DQN (Top) and $\kappa$-VI-DQN (Bottom) on SpaceInvaders, for the hyper-parameter $C_{FA} = 0.05$ (right) and for the 'naive' baseline $N(\kappa) = T$ (left).

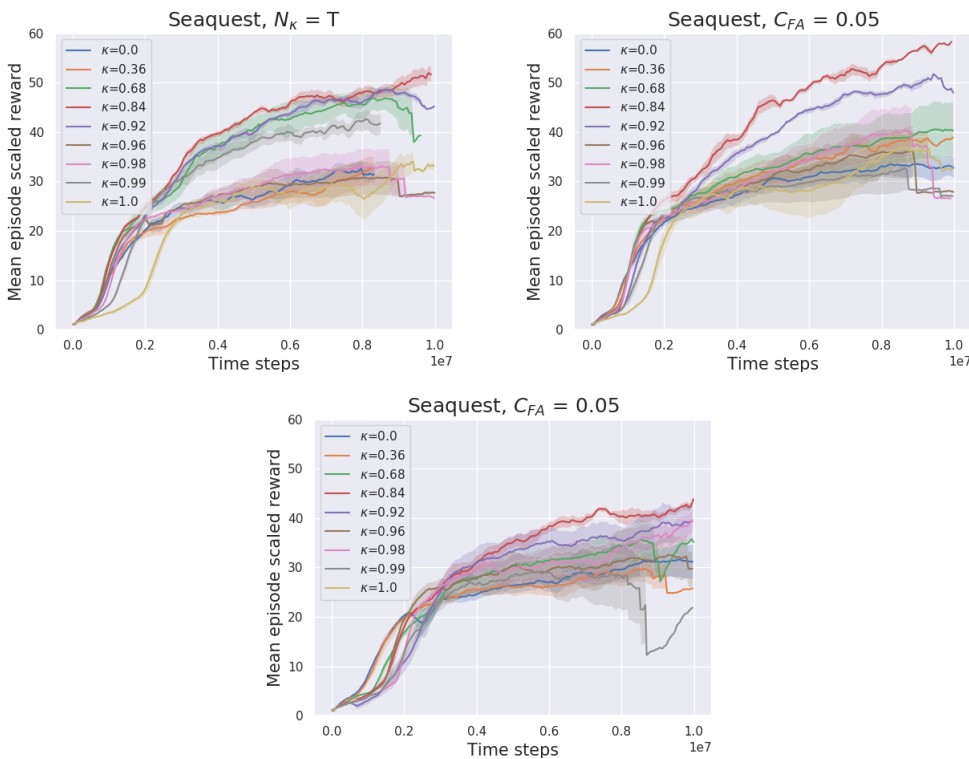

Figure 6: Training performance of $\kappa$-PI-DQN (Top) and $\kappa$-VI-DQN (Bottom) on Seaquest, for the hyper-parameter $C_{FA} = 0.05$ (right) and for the 'naive' baseline $N(\kappa) = T$ (left).

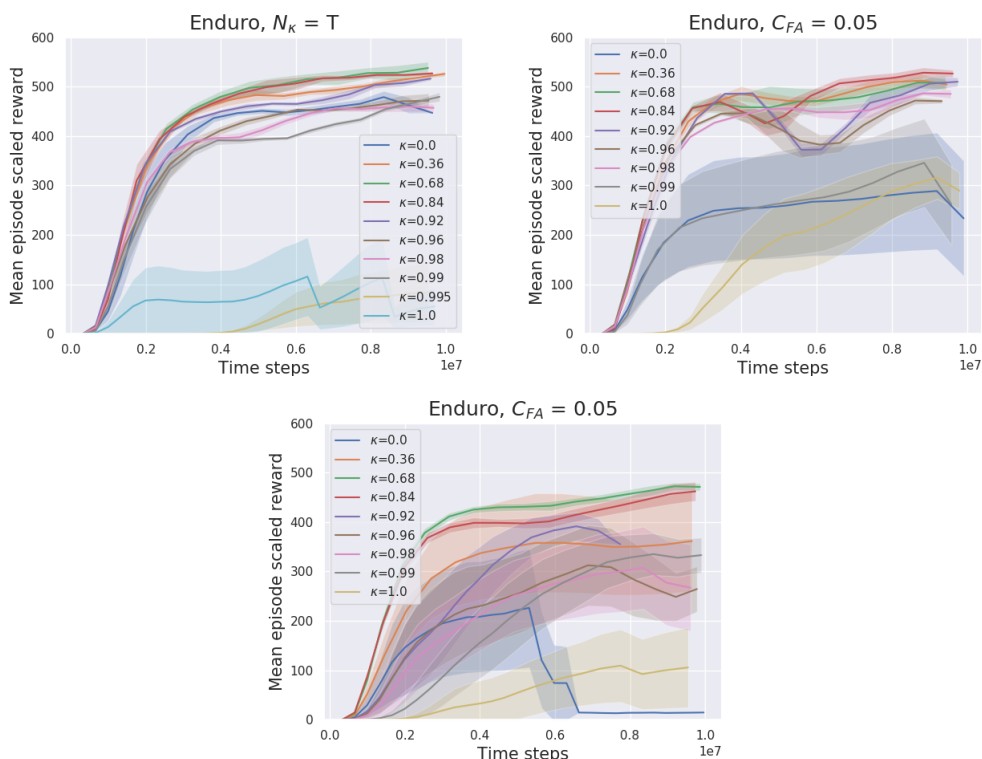

Figure 7: Training performance of $\kappa$-PI-DQN (Top) and $\kappa$-VI-DQN (Bottom) on Enduro, for the hyper-parameter $C_{FA} = 0.05$ (right) and for the 'naive' baseline $N(\kappa) = T$ (left).

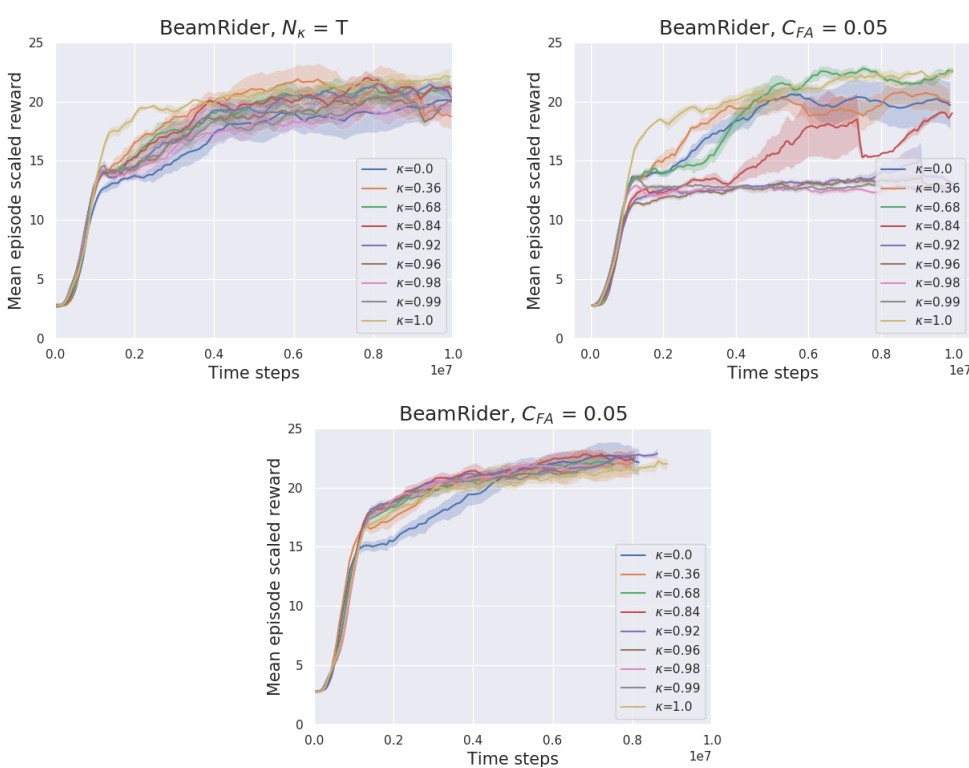

Figure 8: Training performance of $\kappa$-PI-DQN (Top) and $\kappa$-VI-DQN (Bottom) on BeamRider, for the hyper-parameter $C_{FA} = 0.05$ (right) and for the 'naive' baseline $N(\kappa) = T$ (left).

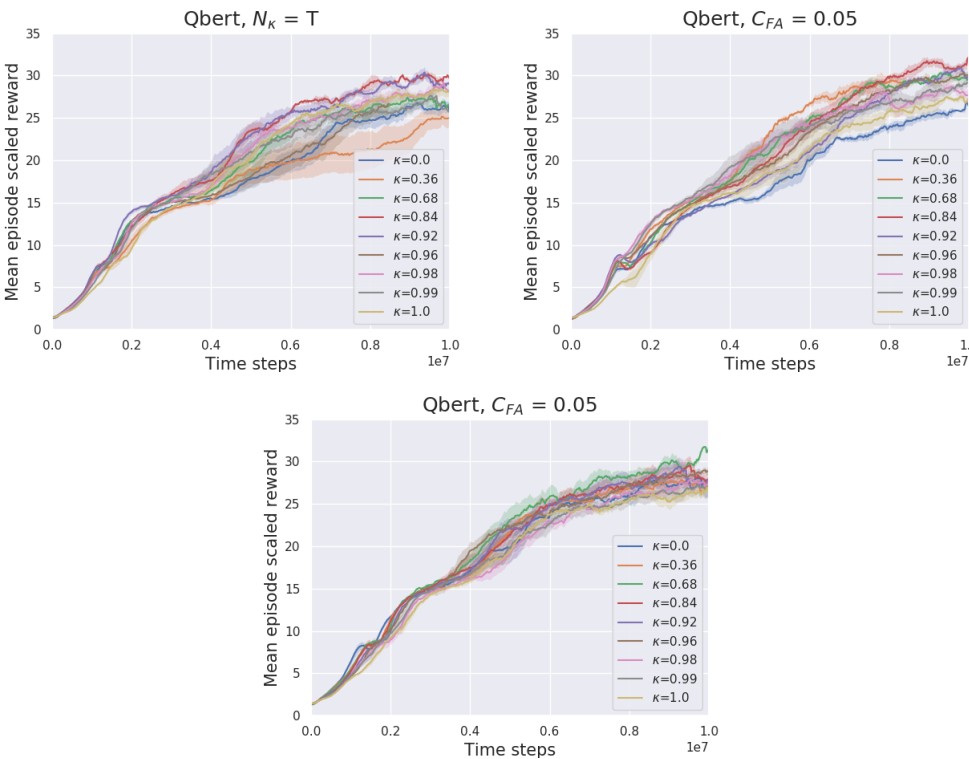

Figure 9: Training performance of $\kappa$-PI-DQN (Top) and $\kappa$-VI-DQN (Bottom) on Qbert, for the hyper-parameter $C_{FA} = 0.05$ (right) and for the 'naive' baseline $N(\kappa) = T$ (left).

## B    TRPO IMPLEMENTATION OF $\kappa$-PI AND $\kappa$-VI

### B.1    PSEUDO-CODES

In this section, we report the detailed pseudo-codes of the $\kappa$-PI-TRPO and $\kappa$-VI-TRPO algorithms, described in Section 5.2, side-by-side.

---

**Algorithm 7** $\kappa$-PI-TRPO

---

1: **Initialize** $V$-networks $V_\theta$ and $V_\phi$, and policy network $\pi_\psi$ with random weights $\theta$, $\phi$, and $\psi$
2: **Initialize** target network $V'_\phi$ with weights $\phi' \leftarrow \phi$
3: **for** $i = 0, \ldots, N(\kappa) - 1$ **do**
4:    **for** $t = 1, \ldots, T(\kappa)$ **do**
5:       Simulate the current policy $\pi_\psi$ for $M$ time-steps;
6:       **for** $j = 1, \ldots, M$ **do**
7:          Calculate $R_j(\kappa, V'_\phi) = \sum_{t=j}^{M} (\gamma\kappa)^{t-j} r_t(\kappa, V'_\phi)$    and    $\rho_j = \sum_{t=j}^{M} \gamma^{t-j} r_t$;
8:       **end for**
9:       Sample a random mini-batch $\{(s_j, a_j, r_j, s_{j+1})\}_{j=1}^{N}$ from the simulated $M$ time-steps;
10:       Update $\theta$ by minimizing the loss function:    $\mathcal{L}_{V_\theta} = \frac{1}{N} \sum_{j=1}^{N} (V_\theta(s_j) - R_j(\kappa, V'_\phi))^2$;
11:       # Policy Improvement
12:       Sample a random mini-batch $\{(s_j, a_j, r_j, s_{j+1})\}_{j=1}^{N}$ from the simulated $M$ time-steps;
13:       Update $\psi$ using TRPO with advantage function computed by $\{(R_j(\kappa, V'_\phi), V_\theta(s_j))\}_{j=1}^{N}$;
14:       # Policy Evaluation
15:       Sample a random mini-batch $\{(s_j, a_j, r_j, s_{j+1})\}_{j=1}^{N}$ from the simulated $M$ time-steps;
16:       Update $\phi$ by minimizing the loss function:    $\mathcal{L}_{V_\phi} = \frac{1}{N} \sum_{j=1}^{N} (V_\phi(s_j) - \rho_j)^2$;
17:    **end for**
18:    Copy $\phi$ to $\phi'$    $(\phi' \leftarrow \phi)$;
19: **end for**

---

---

**Algorithm 8** $\kappa$-VI-TRPO

---

1: **Initialize** $V$-network $V_\theta$ and policy network $\pi_\psi$ with random weights $\theta$ and $\psi$;
2: **Initialize** target network $V'_\phi$ with weights $\phi'; \leftarrow \theta$;
3: **for** $i = 0, \ldots, N(\kappa) - 1$ **do**
4:    # Evaluate $T_\kappa V'_\phi$ and the $\kappa$-greedy policy w.r.t. $V'_\phi$
5:    **for** $t = 1, \ldots, T(\kappa)$ **do**
6:       Simulate the current policy $\pi_\psi$ for $M$ time-steps;
7:       **for** $j = 1, \ldots, M$ **do**
8:          Calculate $R_j(\kappa, V'_\phi) = \sum_{t=j}^{M} (\gamma\kappa)^{t-j} r_t(\kappa, V'_\phi)$;
9:       **end for**
10:       Sample a random mini-batch $\{(s_j, a_j, r_j, s_{j+1})\}_{j=1}^{N}$ from the simulated $M$ time-steps
11:       Update $\theta$ by minimizing the loss function:    $\mathcal{L}_{V_\theta} = \frac{1}{N} \sum_{j=1}^{N} (V_\theta(s_j) - R_j(\kappa, V'_\phi))^2$;
12:       Sample a random mini-batch $\{(s_j, a_j, r_j, s_{j+1})\}_{j=1}^{N}$ from the simulated $M$ time-steps
13:       Update $\psi$ using TRPO with advantage function computed by $\{(R_j(\kappa, V'_\phi), V_\theta(s_j))\}_{j=1}^{N}$;
14:    **end for**
15:    Copy $\theta$ to $\phi'$    $(\phi' \leftarrow \theta)$;
16: **end for**

---

| Hyperparameter | Value |
|---|---|
| Horizon (T) | 1000 |
| Adam stepsize | $1 \times 10^{-3}$ |
| Number of samples per Iteration | 1024 |
| Entropy coefficient | 0.01 |
| Discount factor | 0.99 |
| Number of Iterations | 2000 |
| Minibatch size | 128 |
| #Runs used for plot averages | 5 |
| Confidence interval for plot runs | $\sim 70\%$ |

Table 4: Hyper-parameters of $\kappa$-PI-TRPO and $\kappa$-VI-TRPO on the MuJoCo domains.

## B.2 ABLATION TEST FOR $C_{FA}$

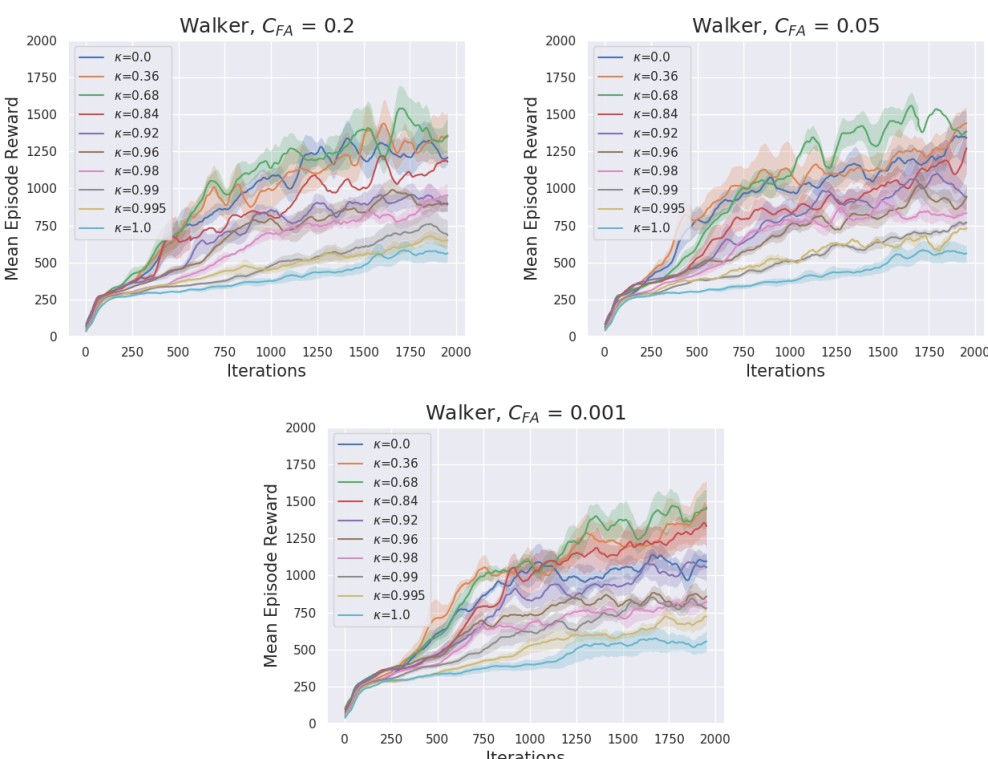

Figure 10: Performance of $\kappa$-PI-TRPO and $\kappa$-VI-TRPO on Walker2d-v2 for different values of $C_{FA}$.

## B.3 $\kappa$-PI-TRPO AND $\kappa$-VI-TRPO PLOTS

In this section, we report additional results of the application of $\kappa$-PI-TRPO and $\kappa$-VI-TRPO on the MuJoCo domains. A summary of these results has been reported in Table 2 in the main paper.

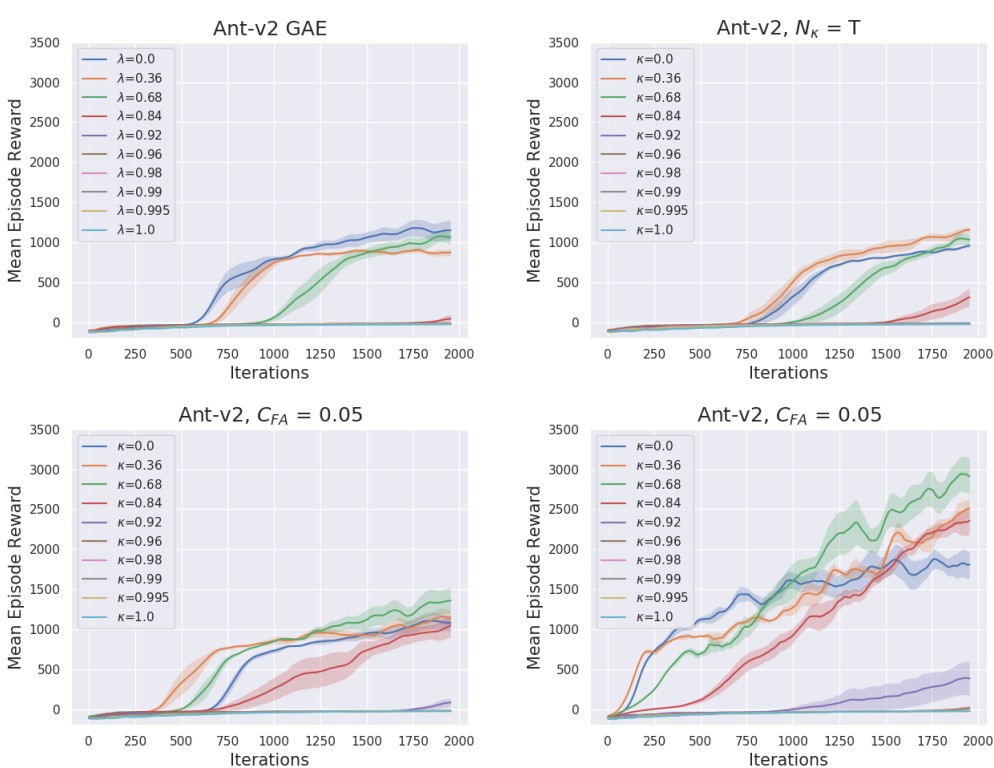

Figure 11: Performance of GAE and 'Naive' baseline (Top row) and $\kappa$-PI-TRPO (Bottom left) and $\kappa$-VI-TRPO (Bottom right) on Ant-v2.

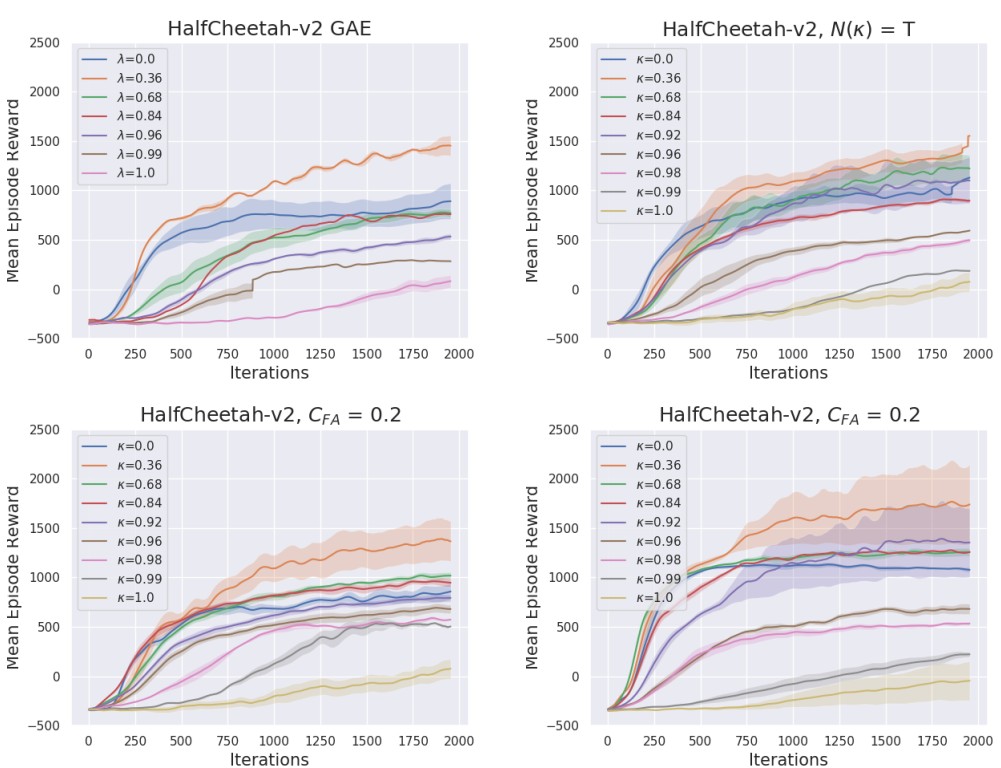

Figure 12: Performance of GAE and 'Naive' baseline (Top row) and $\kappa$-PI-TRPO (Bottom left) and $\kappa$-VI-TRPO (Bottom right) on HalfCheetah-v2.

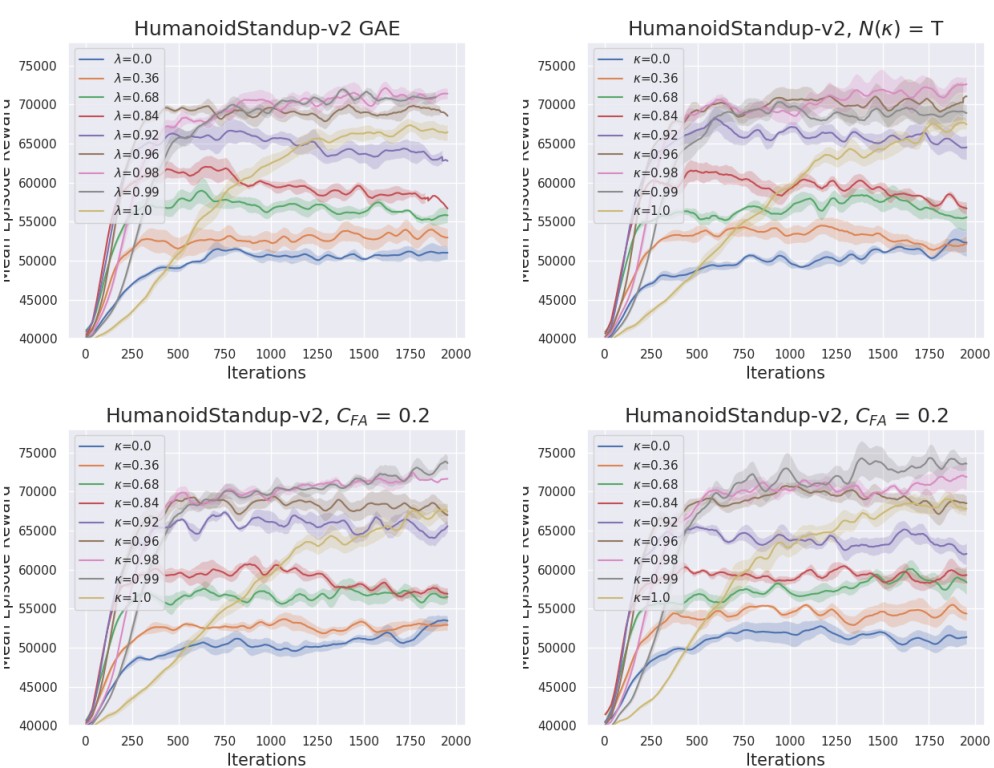

Figure 13: Performance of GAE and 'Naive' baseline (Top row) and $\kappa$-PI-TRPO (Bottom left) and $\kappa$-VI-TRPO (Bottom right) on HumanoidStandup-v2.

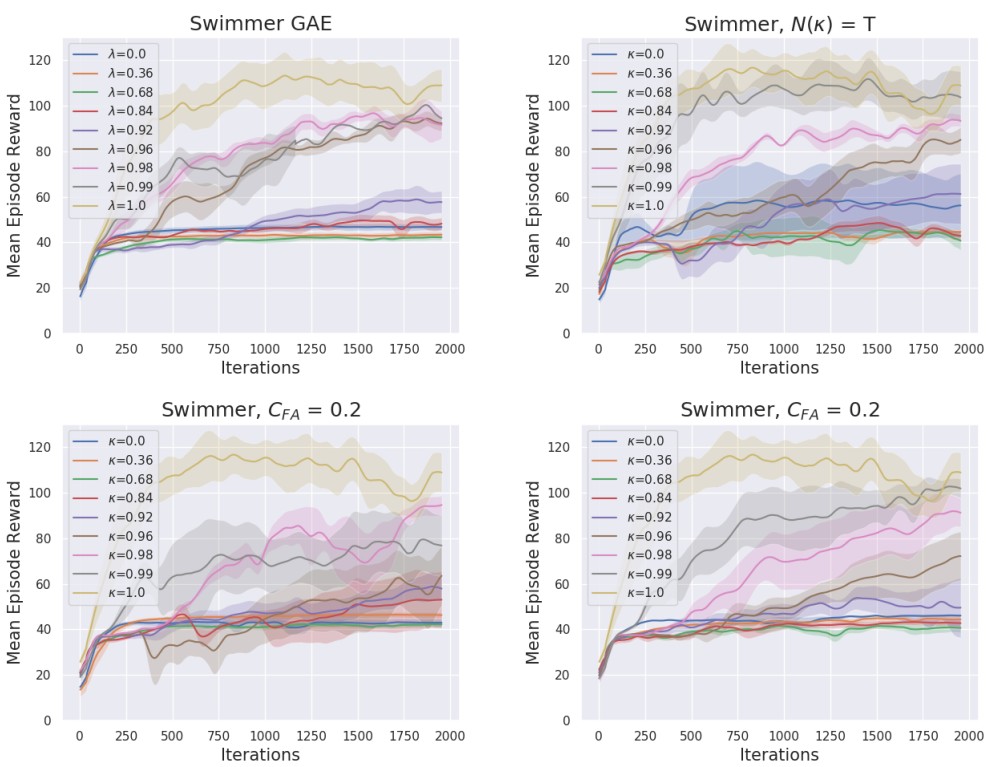

Figure 14: Performance of GAE and 'Naive' baseline (Top row) and $\kappa$-PI-TRPO (Bottom left) and $\kappa$-VI-TRPO (Bottom right) on Swimmer-v2.

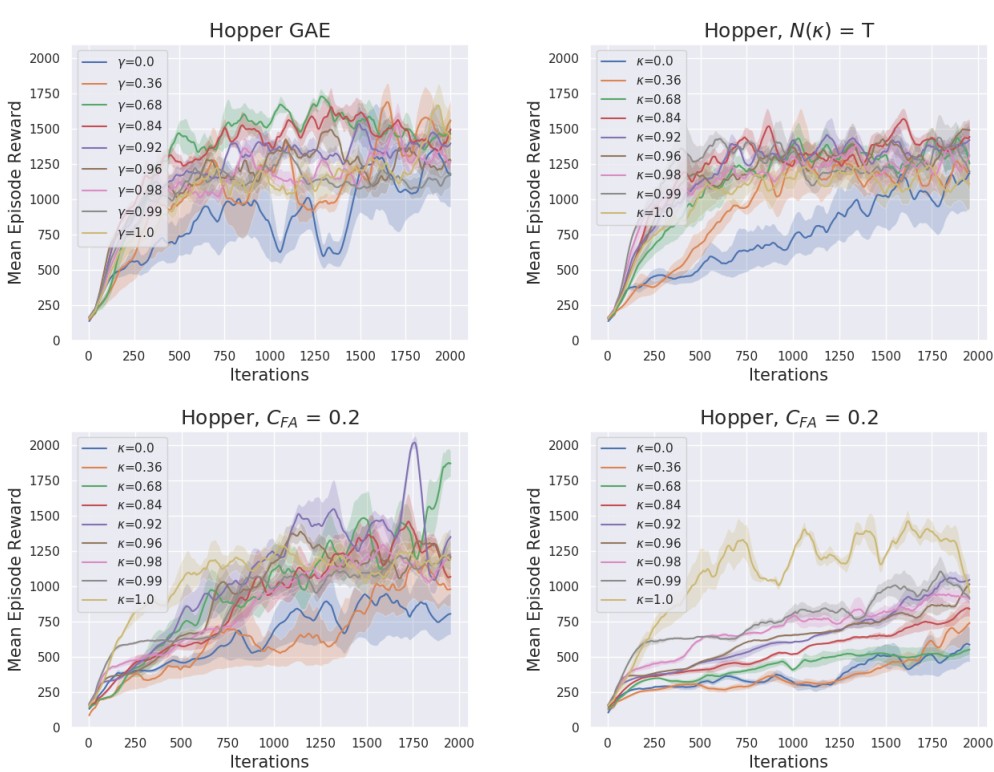

Figure 15: Performance of GAE and 'Naive' baseline (Top row) and $\kappa$-PI-TRPO (Bottom left) and $\kappa$-VI-TRPO (Bottom right) on Hopper-v2.

## C    REBUTTAL RESULTS

### C.1    CARTPOLE

In this section, we analyze the role $\kappa$ plays in the proposed methods by reporting results on the simple CartPole environment for $\kappa$-PI TRPO. For all experiments, we use a single layered value function network and a linear policy network. Each hyperparameter configuration is run for 10 different random seeds and plots are shown for a 50% confidence interval.

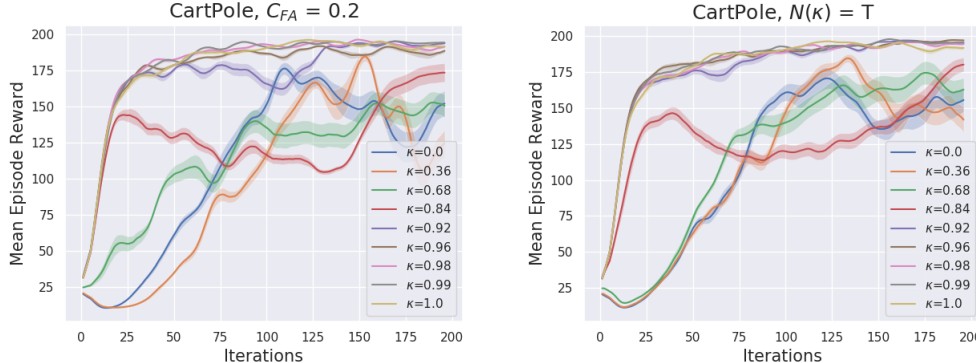

Figure 16: Training performance of $\kappa$-PI-TRPO (Left) and the 'naive' baseline $N(\kappa) = T$ (Right) on the CartPole environment.

Note that since the CartPole is extremely simple, we do not see a clear difference between the $\kappa$ values that are closer to 1.0 (see Figure 16). Below, we observe the performance when the discount factor $\gamma$ is lowered (see Figure 17). Since, there is a ceiling of $R = 200$ on the maximum achievable return, it makes intuitive sense that observing the $\kappa$ effect for a lower gamma value such as $\gamma = 0.36$ will allow us to see a clearer trade-off between $\kappa$ values. To this end, we also plot the results for when the discount factor is set to 0.36.

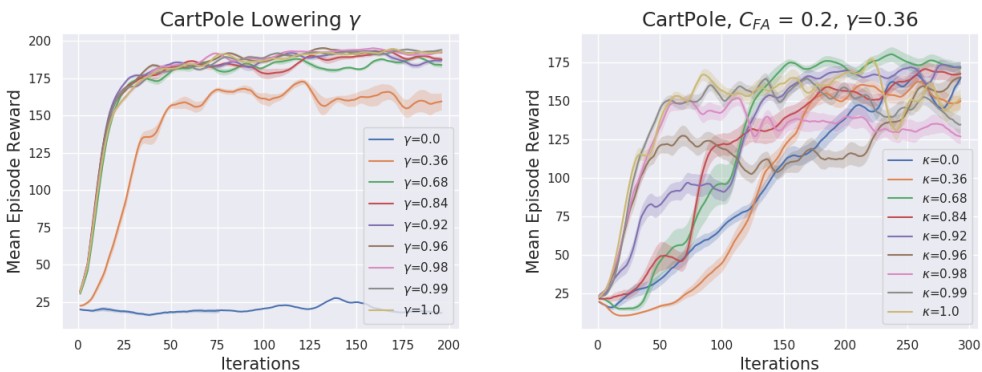

Figure 17: Training performance when lowering the discount factor (Left) and of $\kappa$-PI-TRPO (Left) when discount factor is set to 0.36 (Right) on the CartPole environment.

The intuitive idea behind $\kappa$-PI, and $\kappa$-VI similarly, is that at every time step, we wish to solve a simpler sub-problem, i.e. the $\gamma\kappa$ discounted MDP. Although, we are solving an easier/shorter horizon problem, in doing so, the bias induced is taken care of by the modified reward in this new MDP. Therefore, it becomes interesting to look at how $\kappa$ affects its two contributions, one being the discounting, the other being the weighting of the shaped reward (see eq. 11). Below we look at what happens when each of these terms are made $\kappa$ independent, one at a time, while varying $\kappa$ for the other term. To make this clear, we introduce different notations for both such $\kappa$ instances, one being $\kappa_d$ (responsible for discounting) and the other being $\kappa_s$ (responsible for shaping).

$$\pi_\kappa(s) \in \arg\max_\pi \mathbb{E}[\sum_{t \geq 0} \underbrace{(\gamma \kappa_d)^t}_{\text{discounting}} [r(s_t, a_t) + \underbrace{(1 - \kappa_s)\gamma V(s_{t+1})}_{\text{shaping}}] \mid s_0 = s, \pi], \quad \forall s \in \mathcal{S} \quad (11)$$

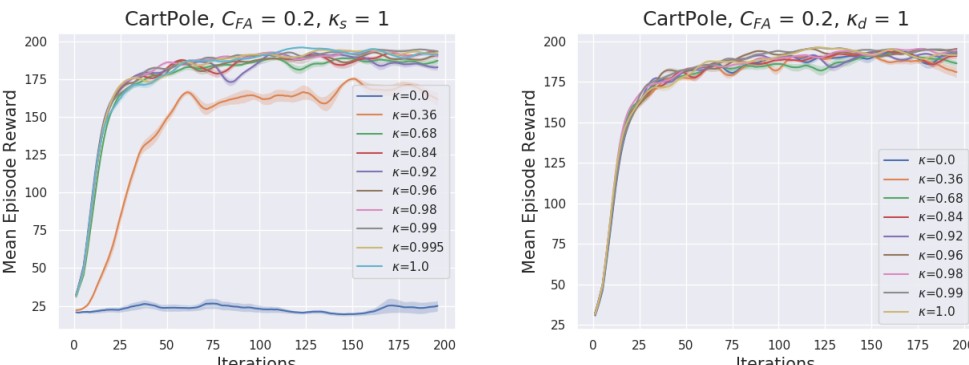

Figure 18: Training performance when contribution of $\kappa$ is fixed for a) the shaped reward (Left) and b) the discount factor (Right) on the CartPole environment.

We see something interesting here. For the CartPole domain, the shaping term does not seem to have any effect on the performance (Figure 18(b)), while the discounting term does. This implies that the problem does not suffer from any bias issues. Thus, the correction provided by the shaped term is not needed. However, this is not true for other more complex problems. This is also why we see a similar result when lowering $\gamma$ in this case, but not for more complex problems.

## C.2 MOUNTAIN CAR

In this section, we report results for the Mountain Car environment. Contrary to the CartPole results, where lowering the $\kappa$ values degraded the performance, we observe that performance deteriorates when $\kappa$ is increased. We also plot a bar graph, with the cumulative score on the y axis and different $\kappa$ values on the x axis.

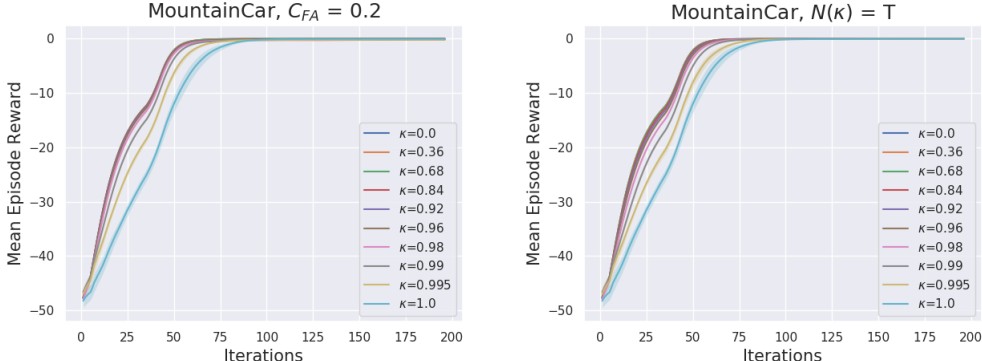

Figure 19: Training performance of $\kappa$-PI-TRPO (Top Left: training curves, Top Right: cumulative return) and the 'naive' baseline $N(\kappa) = T$ (Bottom) on the Mountain Car environment.

We use the continuous Mountain Car domain here, which has been shown to create exploration issues. Therefore, without receiving any positive reward, using a $\kappa$ value of 0 in the case of discounting (solving the 1 step problem has the least negative reward) and of 1 in the case of shaping results in the best performance.

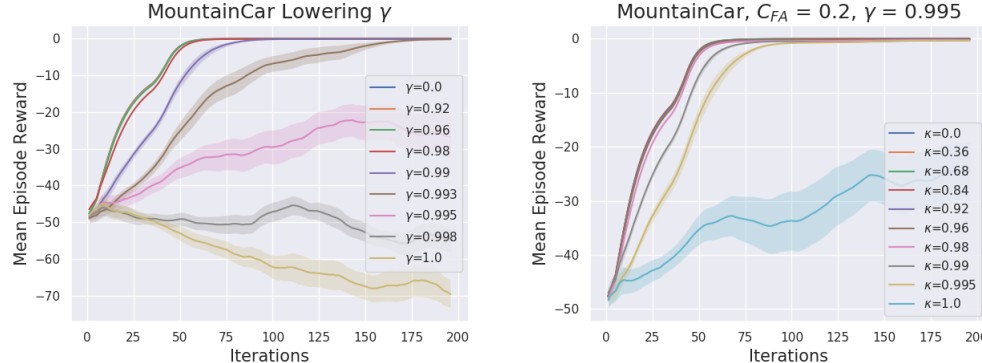

Figure 20: Training performance when lowering the discount factor (Left) and of $\kappa$-PI-TRPO (Left) when discount factor is set to 0.995 (Right) on the MountainCar environment.

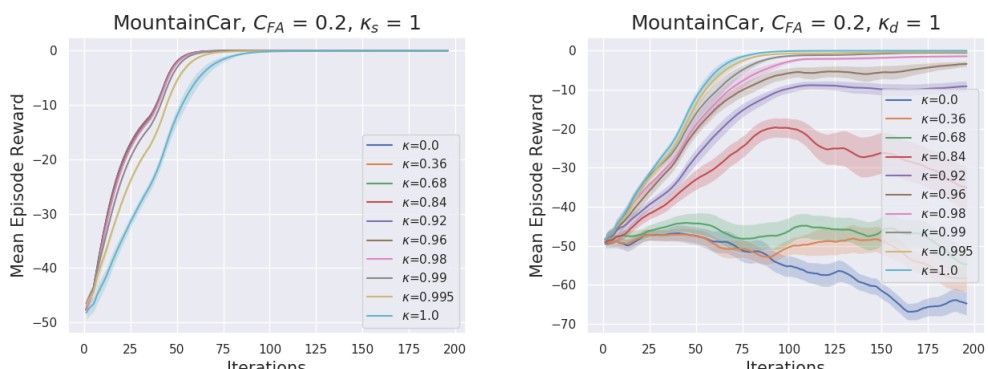

Figure 21: Training performance when contribution of $\kappa$ is fixed for a) the shaped reward (Left) and b) the discount factor (Right) on the MountainCar environment.

### C.3   PENDULUM

In this section, we move to the Pendulum environment, a domain where we see a non-trivial best $\kappa$ value. This is due to there not being a ceiling on the maximum possible return, which is the case in CartPole.

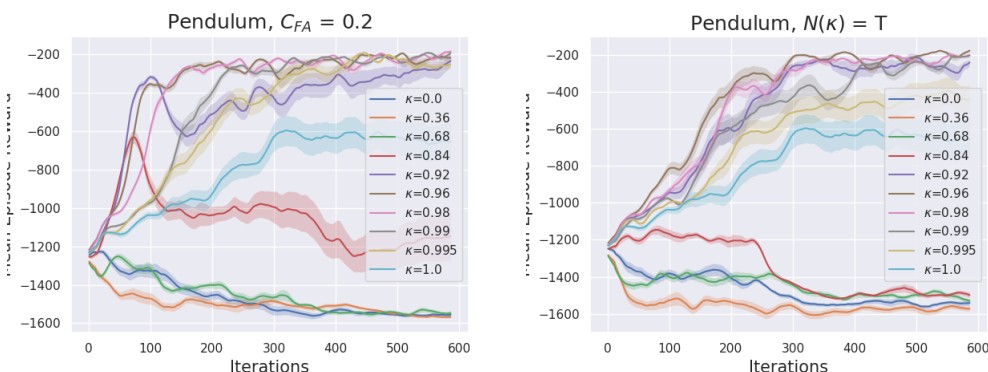

Figure 22: Training performance of $\kappa$-PI-TRPO (Top Left: training curves, Top Right: cumulative return) and the 'naive' baseline $N(\kappa) = T$ (Bottom) on the Pendulum environment.

Choosing the best $\gamma$ value and running $\kappa$-PI on it results in an improved performance for all $\kappa$ values (see Figure 23).

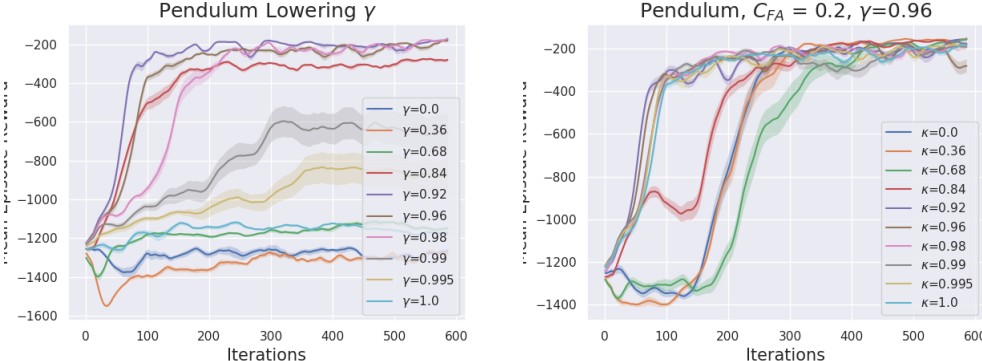

Figure 23: Training performance when lowering the discount factor (Left) and of $\kappa$-PI-TRPO (Left) when discount factor is set to 0.96 (Right) on the Pendulum environment.

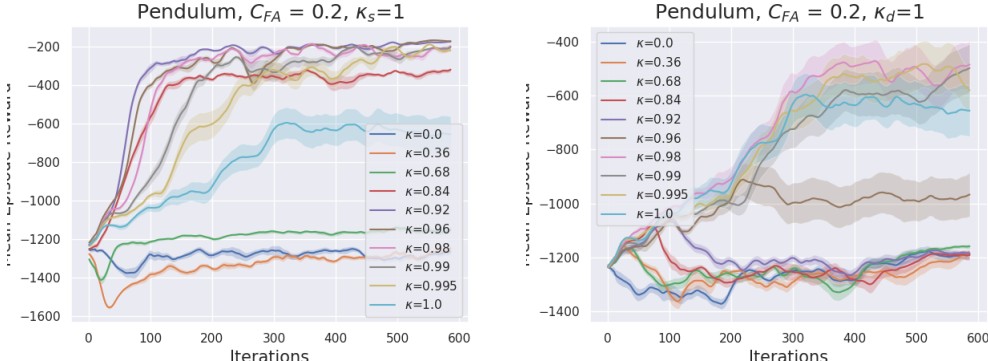

Figure 24: Training performance when contribution of $\kappa$ is fixed for a) the shaped reward (Left) and b) the discount factor (Right) on the Pendulum environment.

To summarize, we believe that in inherently short horizon domains (dense, per time step reward), such as the Mujoco continuous control tasks, the discounting produced by $\kappa$-PI and VI is shown to cause major improvement in performance over the TRPO baselines. This is reinforced by the results of lowering the discount factor experiments. On the other hand, in inherently long horizon domains (sparse, end of trajectory reward), such as in Atari, the shaping produced by $\kappa$-PI and VI is supposed to cause the major improvement over the DQN baselines. Again, this is supported by the fact that lowering the discount factor experiments actually result in deterioration in performance.

## C.4 SMOOTHNESS IN $\kappa$

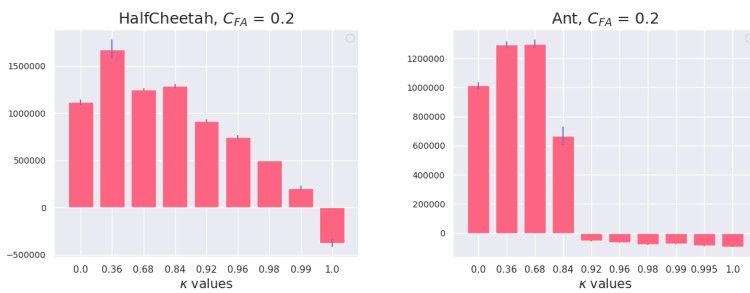

Figure 25: Cumulative training performance of $\kappa$-PI-TRPO on HalfCheetah (Left, corresponds to Figure 12) and Ant (Right, corresponds to Figure 11) environments.

