# OpenReview forum: "Multi-step Greedy Policies in Model-Free Deep Reinforcement Learning"
_ICLR.cc/2020/Conference — Reject_

### Official Review · AnonReviewer1 · 2019-10-11
**Official Blind Review #1**

**Rating:** 3

**Review:**

This paper focuses on the implementation and some empirical evaluations of a class of algorithms designed to find optimal strategies/values of large MDP.

The basic idea of these algorithm (called \kappa-PI or \kappa-VI) is to combine two type of classical approaches:
- policy/value iteration
- k-step ahead computation (instead of just 1-ahead, and actually, k should be quite big or even infinite with an auxiliary appropriate discount rate).

The theoretical formulation of \kappa-PI and \kappa-VI involves solving, at each iteration, another auxiliary MDP problem (where the discount rate is of order \kappa\gamma). This is basically what this paper suggests to do, and implements.

The experiments are a bit difficult for me to read, as the baselines (\kappa=0 and =1, say) are compared with "the best \kappa" which seems to be problem dependent, so I do not know if there is a clear message.

**Experience Assessment:**

I do not know much about this area.

**Review Assessment: Checking Correctness Of Derivations And Theory:**

I did not assess the derivations or theory.

**Review Assessment: Checking Correctness Of Experiments:**

I did not assess the experiments.

**Review Assessment: Thoroughness In Paper Reading:**

I read the paper at least twice and used my best judgement in assessing the paper.

---

> ### Author Response · Authors · 2019-11-09
> **Response to AnonReviewer1 (2/2)**
>
> *****Response to the Reviewer’s Comments*****
> At the beginning, we would like to bring it to the reviewer’s attention that $\kappa$ is a parameter in the range [0,1] and cannot go to infinity. $\kappa$=0 corresponds to 1-step greedy and $\kappa$=1 corresponds to solving the entire MDP. This fact makes the resulting algorithms easy to implement, unlike an approach that uses finite lookahead policies.
>
> We now provide a summary of our experiments and the lessons one can learn from them. Hope this helps the reviewer with reading the experiments, and clarifies the messages we would like to deliver in this work.
>
> The goal behind our experiments is to compare against the DQN and TRPO baselines, which are special cases of our algorithm by setting $\kappa$=1.
>
> The first takeaway message is that there are non-trivial $\kappa$ values for which we could observe better performance than DQN and TRPO. These $\kappa$ values are different for different environments. Here are some results revealed through our work:
>
>     - We can categorize each environment with a certain range of ‘ideal’ $\kappa$ values, e.g., either lower or higher $\kappa$ values.
>
>     - Our results also show that in TRPO, although previous work, such as GAE, concluded to have a fixed $\lambda$ parameter across all environments, this is certainly not true. A $\kappa$ or a $\lambda$ value that works well for one environment is not guaranteed to be working well for another. Therefore, the natural next step, that we are currently working on,  is to build methods that can adapt the value of $\kappa$ based on the problem at hand.
>
> Secondly, since our methods have been derived from Policy/Value Iteration schemes, it makes sense to check how well they work when the policy evaluation and improvement steps are separated, i.e., improving for multiple time steps before evaluating the policy. We do this through the ‘naive’ baseline comparison which improves the policy for a single time-step. The results consistently show that doing a multi-step update is better. This is the second takeaway message from our work.
>
> Thirdly, one can also wonder what effect does lowering the discount factor have on the problem, since the $\kappa$-PI algorithm advocates for solving a more discounted MDP (i.e., the $\gamma\kappa$ MDP, instead of the $\gamma$ MDP, at each time step). Our results show that the comparison is non-trivial, as we achieve consistently better performance with $\kappa$ PI/VI, while lowering the discount factor actually hurts the baseline performance in most cases. This forms the third take-away of our work.

---

> ### Author Response · Authors · 2019-11-09
> **Response to AnonReviewer1 (1/2)**
>
> We would like to thank the reviewer for the comments.
>
> Before addressing the reviewer’s comments, we would like to summarize the contributions of our work.
>
> *****Summary*****
> The advantage of lowering the discount factor in RL has been investigated in several prior work, e.g., Petrik and Scherrer (2009) and Jiang et al. (2015). However, they have shown that lowering the discount factor introduces bias, and as we empirically demonstrate in Section 5.3, this bias can lead to a deterioration in the performance.
>
> In our work, instead of lowering the discount factor, we follow a different route that was theoretically formulated in Efroni et al. (2018a). They introduced the notion of $\kappa$ greedy policy, from which they derived $\kappa$-Policy Iteration ($\kappa$-PI) and $\kappa$-Value Iteration ($\kappa$-VI) algorithms. In their previous work (Efroni et al., 2018a and 2018b), they only theoretically analyzed these algorithms and empirically evaluated their convergence speed in problems with small number of states (100 states at most) and where the model of the environment is known (planning setting, not learning setting). It is not obvious if their theoretical results apply when complex function approximations, such as deep neural networks, are used to solve the problem in the model-free setting (learning, not planning). In this work, we investigate extending the algorithms proposed by Efroni et al (2018a,b) to model-free and function approximation settings with both discrete and continuous actions, and show that this extension is non-trivial (as also pointed out by Reviewer 3) and care should be taken in deriving the practical versions of these algorithms (e.g., the importance of the C_{FA} parameter). Furthermore, we demonstrate the generality of the framework by showing that popular algorithms, DQN and TRPO, are special cases of our multi-step greedy framework for $\kappa$ = 1.
>
> We show the advantage of using $\kappa$-PI and $\kappa$-VI algorithms over lowering the discount factor for both value (DQN) and policy (TRPO) based algorithms, when neural networks are used as function approximator (as mentioned in the first paragraph). Our results (in Section 5.3) indicate that while the performance of DQN and TRPO degrades with lowering the discount factor, our multi-step greedy algorithms improve over DQN and TRPO.
>
> Furthermore, we test some of the consequences of the theory in Efroni et al. (2018a,b) on multi-step greedy dynamic programming. In particular, we show the advantage of using ‘hard’ updates over ‘soft’ updates, which was shown to be problematic (theoretically) by Efroni et al. (2018b). By hard and soft updates (terms used in Efroni et al., 2018b), we refer to fully solving the $\gamma\kappa$ MDP in a model-free manner (hard) versus changing the policy at each iteration (soft). In the ‘hard’ setting, the policy improvement and evaluation steps are separated, while in the ‘soft’ setting, they are concurrent (each policy improvement step is followed by a policy evaluation step).
>
> We believe much more is left to be understood in applying multi-step greedy policies to RL. We consider our work as a first step towards this goal as well as showing such an approach is empirically beneficial.

---

### Official Review · AnonReviewer2 · 2019-10-22
**Official Blind Review #2**

**Rating:** 3

**Review:**

The main contributions of this paper are k-PI-DQN and k-VI-DQN, which are model-free versions of dynamic programming (DP) methods k-PI and k-VI from another paper (Efroni et al., 2018).  The deep architecture of the two algorithms follows that of DQN.  Efroni et al. (2018b) already gave a stochastic online (model-free) version of k-PI in the tabular setting.  Although this paper is going one step further extending from tabular to function approximation, I feel that the paper just combined known results, the shaped reward from Efroni et al (2018a) and DQN.  The extension seems straightforward.  Mentioning previous results from Efroni et al (2018a) and (2018b) does not justify the extension would possess the same property or behaviour.   The experiments were only comparing their methods with different hyperparameters, with only a brief comparison to DQN.

**Experience Assessment:**

I have read many papers in this area.

**Review Assessment: Checking Correctness Of Derivations And Theory:**

I assessed the sensibility of the derivations and theory.

**Review Assessment: Checking Correctness Of Experiments:**

I assessed the sensibility of the experiments.

**Review Assessment: Thoroughness In Paper Reading:**

I read the paper at least twice and used my best judgement in assessing the paper.

---

> ### Author Response · Authors · 2019-11-09
> **Response to AnonReviewer2 (2/2)**
>
> “no mention of our TRPO work in the review”
> The review only mentions our DQN work and does not talk about our TRPO algorithms and experiments. We are sorry if we did not present this part of our work clearly enough. We will improve the presentation of this part in the final version of the paper. To clarify, we experimented with both DQN and TRPO extensions of our approach. As stated in the paper, the TRPO extension resembles the practically used GAE (Generalized Advantage Estimation) algorithm, with a crucial difference that in GAE the value and policy are concurrently updated (each policy improvement step is followed by a policy evaluation step), while in our work, we emphasize the need to do multiple step improvement before evaluating the policy. The theoretical results of Efroni et al. (2018b) suggest that the concurrent update approach used by GAE does not necessarily result in an improving algorithm, which hints that using this approach might be problematic. We conjecture that the reason that this issue does not lead to significant performance deterioration in GAE is that most MuJoCo continuous control tasks are inherently of short horizon. In fact, our experiments show that in the Atari domains concurrently learning the policy and value leads to inferior performance.

---

> ### Author Response · Authors · 2019-11-09
> **Response to AnonReviewer2 (1/2)**
>
> We would like to thank the reviewer for the comments.
>
> Before addressing the reviewer’s comments, we would like to summarize the contributions of our work.
>
> *****Summary*****
> The advantage of lowering the discount factor in RL has been investigated in several prior work, e.g., Petrik and Scherrer (2009) and Jiang et al. (2015). However, they have shown that lowering the discount factor introduces bias, and as we empirically demonstrate in Section 5.3, this bias can lead to a deterioration in the performance.
>
> In our work, instead of lowering the discount factor, we follow a different route that was theoretically formulated in Efroni et al. (2018a). They introduced the notion of $\kappa$ greedy policy, from which they derived $\kappa$-Policy Iteration ($\kappa$-PI) and $\kappa$-Value Iteration ($\kappa$-VI) algorithms. In their previous work (Efroni et al., 2018a and 2018b), they only theoretically analyzed these algorithms and empirically evaluated their convergence speed in problems with small number of states (100 states at most) and where the model of the environment is known (planning setting, not learning setting). It is not obvious if their theoretical results apply when complex function approximations, such as deep neural networks, are used to solve the problem in the model-free setting (learning, not planning). In this work, we investigate extending the algorithms proposed by Efroni et al (2018a,b) to model-free and function approximation settings with both discrete and continuous actions, and show that this extension is non-trivial (as also pointed out by Reviewer 3) and care should be taken in deriving the practical versions of these algorithms (e.g., the importance of the C_{FA} parameter). Furthermore, we demonstrate the generality of the framework by showing that popular algorithms, DQN and TRPO, are special cases of our multi-step greedy framework for $\kappa$ = 1.
>
> We show the advantage of using $\kappa$-PI and $\kappa$-VI algorithms over lowering the discount factor for both value (DQN) and policy (TRPO) based algorithms, when neural networks are used as function approximator (as mentioned in the first paragraph). Our results (in Section 5.3) indicate that while the performance of DQN and TRPO degrades with lowering the discount factor, our multi-step greedy algorithms improve over DQN and TRPO.
>
> Furthermore, we test some of the consequences of the theory in Efroni et al. (2018a,b) on multi-step greedy dynamic programming. In particular, we show the advantage of using ‘hard’ updates over ‘soft’ updates, which was shown to be problematic (theoretically) by Efroni et al. (2018b). By hard and soft updates (terms used in Efroni et al., 2018b), we refer to fully solving the $\gamma\kappa$ MDP in a model-free manner (hard) versus changing the policy at each iteration (soft). In the ‘hard’ setting, the policy improvement and evaluation steps are separated, while in the ‘soft’ setting, they are concurrent (each policy improvement step is followed by a policy evaluation step).
>
> We believe much more is left to be understood in applying multi-step greedy policies to RL. We consider our work as a first step towards this goal as well as showing such an approach is empirically beneficial.
>
> *****Response to the Reviewer’s Comments*****
> “this work extends Efroni et al. (2018b) from tabular to function approximation”
> This reviewer’s statement is not completely accurate. The work of Efroni et al (2018a) and (2018b) focused on theoretical analysis of $\kappa$ PI/VI algorithms, and their experiments are in small (tabular) problems, where the model is given (the setting is planning, not learning). Our work is the first one to use multi-step greedy policies with neural networks as function approximator, in model-free RL (learning and not planning setting).

---

### Official Review · AnonReviewer3 · 2019-10-23
**Official Blind Review #3**

**Rating:** 6

**Review:**

===== Summary =====
The paper proposes an extension of multi-step dynamic programming algorithms from Efroni, Dalal, Scherrer, and Mannor (2018a, 2018b) to the reinforcement learning setting with function approximation. The multi-step dynamic programming algorithms proposed by Efroni et. al. (2018a)  find the solution of the h-step optimal Bellman operator, which applies the maximum over the next h sequence of actions. Moreover, Efroni et. al. (2018a) also showed an equivalence between h-step optimal Bellman operators and k-Policy Iteration (k-PI) and k-Value Iteration (k-VI) algorithms, which, similar to TD( 𝜆 ) but for policy improvement, take a geometric average of all future h-step returns weighted by k. The paper extends the work from Efroni et. al. (2018a, 2018b) to the deep reinforcement learning setting by proposing an approximate k-PI and k-VI algorithm based on DQN and TRPO. Finally, the paper provides empirical evaluations of k-PI and k-VI with DQN in several Atari games and of k-PI and k-VI with TRPO in several MuJoCo environments with continuous actions paces.

Contributions:
1. The paper proposes a non-trivial extension for k-PI and k-VI to use function approximation via the DQN algorithm.
2. Similarly, the paper proposes a non-trivial extension for k-PI and k-VI to use function approximation with continuous action spaces via the TRPO algorithm.
3. The paper provides empirical evaluations of the four proposed algorithms and, at least for the k-PI algorithm with DQN and TRPO, demonstrates an improvement over the baselines.

===== Decision =====
The paper represents a natural next step to the work of Efroni et. al. (2018a, 2018b). The paper extends the applicability of multi-step greedy policies to more complex environments and shows a statistically significant improvement in performance compared to the methods that it builds upon. Additionally, the ideas are presented clearly and incrementally throughout the paper, which makes it flow nicely until the part where k-PI and k-VI DQN and TRPO are introduced. This is my main complaint about the paper, the lack of simple and intuitive understanding about k-PI and k-VI with function approximation due to the complicated architectures associated with DQN and TRPO. For this reason, my rating of the paper is weak accept.

===== Detailed Comments about Decision =====
All of these are comments for which I would consider increasing my score if they were addressed.

=== Empirical Evaluations ===
First, my main complaint is the complicated architectures and complex domains used to gain insights about k-PI and k-VI with function approximation. Big demonstrations in Atari and MuJoCo are important, but in the case of very new algorithms such as these ones, I consider it to be more important to gain insight through small domains that allow us to dig deep into the algorithms. Any small domain that would allow for big sample sizes for ablation and parameter studies would be more insightful than big demonstrations with very small sample sizes. I do not mean to be dismissive about what has been done in the paper, but it would be a great source of insight and a big improvement to what has already been done if a simple demonstration was presented in the paper.

My suggestion would be to use a simple approximation method, such as Tile Coding with linear function approximation, in small a domain such as mountain car. This would allow for a bigger sample size and a parameter study that could provide more insight about the role of the parameters k and C_{FA} on the performance of k-PI and k-VI.

Additionally, one of the claims in the conclusions was never emphasized in the results: “importantly, the performance of the algorithms was shown to be ‘smooth’  in the parameter k.” This was not completely obvious until I spent some time looking closely at the graph. It eventually became clear, but I think a simpler way to emphasize this is to show a plot of the cumulative reward over the whole training period with the values of k on the x-axis. Based on the top right pane of FIgure 1, this type of plot would show a smooth increase from k=0.99 to k=0.68 followed by a smooth decrease from k=0.68 to k=0.

Finally, I have some questions about some of the choices made in the experiments and results sections:

1. Why choose 50% confidence intervals? 50% confidence intervals with a sample size of 4 in the case of DQN and 5 in the case of TRPO is equivalent to multiplying the standard error by a factor of approximately 0.7, which is narrower than using the standard error on its own. Thus, it seems that some of the conclusions would change based on using a 95% confidence interval compared to a 50% confidence interval in Tables 1 and 2. I insist in showing the performance in a small domain with a simple form of function approximation. This would complement the Atari and MuJoCo experiments by showing improvements in performance with a higher confidence.

2. In remark one, it is pointed out that another target network \tilde Q should be used to obtain \pi_{t-1}, but this was not done to reduce the space complexity of the algorithm. How big were the networks that you used for k-PI DQN? If the network was not prohibitively big, why not implement \tilde Q instead of using an alternative that further deviates from the original k-PI algorithm?

3. Line 19 of Algorithm 5 in Appendix A.1 is supposed to be the off-policy TD(0) update. However, it is not clear how this update is off-policy TD(0) since it based on Q and it does not have any importance sampling to correct for the difference in policies. Am I missing something? It seems that it should be off-policy Sarsa(0), but even then it would still be missing an importance sampling term (see Sutton & Barto, 2018, Equation 7.11, or Algorithm 1 of Precup, Sutton, and Singh, 2000, for more information).

=== Contradictory Claims in the Results ===
There are a few claims that contradict with what is shown in Table 1 and 2.

In the last paragraph of Section 5.1.1 it says that “[the table 1] show[s] that setting N(k) = T leads to a clear degradation of the final training performance on all the domains except Enduro.” This is only true in two out of four games presented in Table 1. In Seaquest the lower confidence bound of the performance of k-PI with k=0.68 is 4643, whereas the upper confidence bound of the performance of k-PI with N(k) = T is 4837; the intervals clearly overlap. Similarly, in the game of Enduro, where k-PI with N(k) = T is said to have better performance, the lower confidence bound of k-PI with N(k) =T is 530, whereas for k-PI with k=0.84 the upper confidence bound is 575; again, the confidence intervals overlap. Hence, neither of these two claims are fully justified, and it is certainly not a “clear degradation of the final training performance.”

Similarly, in Section 5.2.2, k-PI is said to have a better performance than N(k) = T based on the results of Table 2. However, similar calculations show that this is only true for the Ant domain.

===== Minor Comments =====
1. I believe there is a typo in the last column of Table 1, it should be a \kappa instead of a 𝜆.

2. In the second paragraph above Equation 7, the convergence of PI and VI are said to converge to the optimal value with linear rate, but the rate of convergence is O( \gamma^N ), i.e., exponential. Similarly, for the k-PI and k-VI their rate of convergence is O( \ksi ( \kappa )^{N( \kappa )} ), which is also exponential.

===== References =====
Precup, Doina; Sutton, Richard S.; and Singh, Satinder, "Eligibility Traces for Off-Policy Policy Evaluation" (2000).ICML '00 Proceedings of the Seventeenth International Conference on Machine Learning. 80.Retrieved fromhttps://scholarworks.umass.edu/cs_faculty_pubs/80

R. Sutton and A. Barto. Reinforcement learning: An introduction. 2018.

Y. Efroni, G. Dalal, B. Scherrer, and S. Mannor. Beyond the one step greedy approach in reinforcement learning. In Proceedings of the 35th International Conference on Machine Learning, 2018a.

Y. Efroni, G. Dalal, B. Scherrer, and S. Mannor. Multiple-step greedy policies in approximate and online reinforcement learning. In Advances in Neural Information Processing Systems, pp. 5238–5247, 2018b.


**Experience Assessment:**

I have published one or two papers in this area.

**Review Assessment: Checking Correctness Of Derivations And Theory:**

I carefully checked the derivations and theory.

**Review Assessment: Checking Correctness Of Experiments:**

I carefully checked the experiments.

**Review Assessment: Thoroughness In Paper Reading:**

I read the paper thoroughly.

---

> ### Author Response · Authors · 2019-11-09
> **Response to AnonReviewer3**
>
> We would like to thank the reviewer for the detailed review and useful comments.
>
> “using simpler domains to better explain the algorithms”
> We agree with the reviewer that it would be better to describe very new algorithms using small/simple domains. The reason that we did not initially include our experiments in simple domains is that the difference between the performance of the algorithms is not very clear in such problems. To address the reviewer’s concern, we have now added a new section to the paper (Appendix C), where we report results on simpler environments, such as CartPole and Mountain Car. For the experiments on CartPole, we focus on the $\kappa$-PI TRPO algorithm solely, since the results for the other versions ($\kappa$-VI TRPO, DQN, and $\kappa$-PI DQN) follow similarly. As pointed out by the reviewer, the purpose of this section is to gather a more intuitive understanding of the algorithms. Please refer to Appendix C in the updated paper for a more detailed discussion.
>
> “a simpler way to emphasize this is to show a plot of the cumulative reward ...”
> We have added a couple of bar plots to address this in Appendix C.4. These correspond to the $\kappa$-PI training plots for HalfCheetah and Ant domains. It is clear from the bar plots that the performance is smooth in $\kappa$.
>
> “1. Why choose 50% confidence intervals?”
> The results reported in Table 1 and 2 are the empirical mean $\pm$ the empirical standard deviation, which for the sample size of 4 or 5 runs is roughly equal to the 95% confidence interval bound. Moreover, in the plots, we show results describing the empirical mean $\pm$ 0.5 * empirical standard deviation, which is, again for the sample size of 4 or 5 runs, roughly equal to a 60% to 70% confidence interval. This is done so that there is less overlap in the graphs and they are more readable. The 50% value actually corresponds to these plots. We apologize for the lack of clarity here and have updated the paper with the correct confidence values. All conclusions are made with respect to the Table data eventually, which remains unchanged and still corresponds to the 95% confidence bound.
>
> “2. How big were the networks that you used for k-PI DQN?”
> The DQN network sizes are the same as used in Mnih et al. [1], i.e., 3 convolutional layers followed by 2 fully connected layers.
>
> “3. Line 19 of Algorithm 5 in Appendix A.1 …”
> The update in Line 9 resembles the expected SARSA update in Van Seijen et al. [2]. Also, this is the exact update as in DDPG (Equation 5 in Lillicrap et al. [3]).
>
> “Contradictory Claims in the Results”
> Our claim is essentially saying that the mean values of the best performing $\kappa$ are consistently better than the mean values of the N_kappa = T baseline. Since the data here corresponds to the 95% confidence interval bound for 4-5 sample runs, increasing the number of sample runs would decrease the width of the 95% confidence interval, which essentially would ensure no overlap between the upper confidence limit of the baseline and the lower confidence limit of the best $\kappa$ value. For example, comparing the two versions for 10 sample runs in the Ant domain, results in the lower confidence limit of the best $\kappa$ value to be around 1230, while the upper confidence limit of the baseline to be around 1180, hence ensuring no overlap. Please note that due to the inherent variability in final training performance because of random seeding, the mean values for both cases, although relatively consistent, are also slightly changed. However, taking more samples always ensures that any x% confidence bound is narrowed.
>
> “Typo in the last column of Table 1”
> Thank you for pointing this out. We have fixed this in the updated version.
>
> “Linear convergence of PI and VI”
> We apologize for the confusion here. Many of the works in the optimization literature refer to such an exponential rate as linear in the parameter N, and we borrowed the same definition. We have fixed this in the updated version.
>
>
> References:
>
> 1. Mnih, Volodymyr, et al. "Human-level control through deep reinforcement learning." Nature 518.7540 (2015): 529.
> 2. Van Seijen, Harm, et al. "A theoretical and empirical analysis of Expected Sarsa." 2009 IEEE Symposium on Adaptive Dynamic Programming and Reinforcement Learning. IEEE, 2009.
> 3. Lillicrap, Timothy P., et al. "Continuous control with deep reinforcement learning." arXiv preprint arXiv:1509.02971 (2015).

---

> > ### Author Response · Authors · 2019-11-15
> > **Update to Appendix C**
> >
> > For completeness, we have added more results (on the Pendulum domain) in the Appendix C section. Moreover, we have tried to summarize our intuition on why $\kappa$-PI and VI work well in different domains at the end of this section. We believe that this addresses the concerns about intuitive understanding using simpler domains, but would welcome additional suggestions for the final version.

---

> > > ### Comment · AnonReviewer3 · 2019-11-15
> > > **Updated review (1/2)**
> > >
> > > Thank you for your reply and for addressing my comments. However, I still have a few concerns.
> > >
> > > - About the simple domains and architectures
> > > My  main concern was less about the big domains and more about  the complicated architectures used for k-VI/PI DQN and TRPO. With big deep neural network architectures there are too many hyperparameters to tune and control for if one wants to study the effect of one particular part of the algorithm. Thus, I suggested to use a less complex function approximator such as a tilecoder (see Sutton & Barto 2018), which only depends on a few architectural choices and eliminates the representation learning problem from the task. Instead, the authors provided experiments in simpler domains such as cartpole and mountain car. I apologize for the confusion. Nevertheless, the results do shed some clarity on the performance of the algorithms, so I still consider it an improvement. I still have a few questions about the results in mountain car.
> > >
> > > First, what implementation of mountain car are you using? Often the first few episodes in mountain car implemented as in Sutton and Barto (2018) have a cumulative reward below -100, but the plots indicate that the lowest cumulative reward obtained is more than -50. Additionally, with respect to the y-axis of Figure 19 (top left pane), is iterations the same as episodes? If you're using a different implementation from the one in Sutton and Barto (2018) please cite your sources.
> > >
> > > Second, what method was used to smooth the plot? From my experience with this environment, even when averaging over 100 runs, the plots still appear very noisy, so it seems surprising to me that the plots are this smooth with a sample size of 10.
> > >
> > > Additionally, it would be useful to also provide results about the behaviour of the algorithms under different values of CFA. I understand that CFA is related to the final accuracy of V_pi. However, since the V_pi is being approximated anyway, perhaps there may be an interesting trade-off between using a lot of samples to compute an accurate estimate or accept a higher error while moving on to the next policy improvement iteration with a slightly better, yet inaccurate, estimate than before.
> > >
> > > - About the size of the network
> > > The size of the network was less important to me than the rationale behind choosing to deviate further from the original k-PI and k-VI algorithms. In Remark 1, it is mentioned that \tilde{ Q }_\theta, a target network that remains unchanged during the policy improvement step, should be used for \pi_{i-1}; however, because of space complexity Q_\theta', which may have changed during the policy improvement step,  is used instead. My concern about this is that it further deviates from the original algorithms and introduces more confounding factors. Since the purpose of the paper is to study the performance of these algorithms when using deep neural networks to estimate V_pi and Q_pi, then I think it would be better if it was as closed to the original algorithms as possible, which seems doable since the sizes of the networks were not exorbitantly big.
> > >
> > > - Expected Sarsa Update
> > > Line 19 resembles more the update for Sarsa more than Expected Sarsa (see Equation 6.7 of Sutton and Barto , 2018 and compare it to Equation 5 from Van Seijen et. al., 2009). Moreover, neither of those two should be called TD(0) since they are different algorithms. Finally, it is not clear how this is an off-policy update if there is no correction between the current policy and policy used to sample the action at the time it was stored in the experience replay buffer. The bottom line is that the statement that the update in Line 19 corresponds to off-policy TD(0) is false and should be corrected.
> > >
> > > - About the contradictory claims
> > > My main point about that comment was that the claim "[Table 1 and Figure 1] ... lead to a clear degradation of final performance," is an overstatement. In Seaquest, Enduro, Beam Rider, and Qbert the confidence intervals of the performance of k-Pi DQN (k best) and N(k) = T overlap. It is true that with more samples the confidence interval will tend to shrink; however, they will shrink around the true mean, not the observed sample average. That means that if the confidence intervals overlap, there exists a chance that the true mean of N(k) = T will be higher than the k-PI DQN. Since there is a chance that N(k) = T has better performance than k-PI DQN (k best), saying that N(k) = T shows a clear degradation of performance seems exaggerated.
> > >
> > > == More typos ==
> > > - Paragraph above Equation (1), 4th line from the bottom, "The algorithms by which an is be solved...." It feels like there is a word missing.
> > >
> > > == References ==
> > > 1. Sutton, R. S., Barto, A. G. (2018 ). Reinforcement Learning: An Introduction. The MIT Press.
> > >
> > > 2.  Van Seijen, Harm, et al. "A theoretical and empirical analysis of Expected Sarsa." 2009 IEEE Symposium on Adaptive Dynamic Programming and Reinforcement Learning. IEEE, 2009.

---

> > > > ### Comment · AnonReviewer3 · 2019-11-15
> > > > **Updated review (2/2)**
> > > >
> > > > Finally, about my rating of the paper, I have decided to keep it the same: weak accept. However, there is still some work to do on the paper, so I would not mind if the current iteration of the paper was rejected.
> > > >
> > > > As reviewer #1 mentioned, the empirical evaluations in the main body of the paper are hard to read because of the overlap in the graphs. Moreover, although the results show a clear improvement over DQN and TRPO, this fact is lost in the amount of information presented in each graph. Personally, I liked the results in mountain car and pendulum because they are a lot cleaner than the results in Atari and Mujoco. You could also consider using less configuration of the parameters with a bigger spread to more clearly emphasize the results.

---

### Decision · Program_Chairs · 2019-12-19

**Decision:**

Reject

**Comment:**

This paper extends recent multi-step dynamic programming algorithms to reinforcement learning with function approximation.  In particular, the paper extends h-step optimal Bellman operators (and associated k-PI and k-VI algorithms) to deep reinforcement learning.  The paper describes new extensions to DQN and TRPO algorithms.  This approach is claimed to reduce the instability of model-free algorithms, and the approach is tested on Atari and Mujoco domains.

The reviewers noticed several limitations of the work.  The reviewers found little theoretical contribution in this work and they were unsatisfied with the empirical contributions.  The reviewers were unconvinced of the strength and clarity of the empirical results with the Atari and Mujoco domains along with the deep learning network architectures.  The reviewers suggested that simpler domains with a simpler function approximation scheme could enable more through experiments and more conclusive results.  The claim in the abstract of addressing the instabilities was also not adequately studied in the paper.

This paper is not ready for publication.  The primary contribution of this work is the empirical evaluation, and the evaluation is not sufficiently clear for the reviewers.